# Anatomy and function of the vertebral column lymphatic network in mice

Laurent Jacob[1], Ligia Simoes Braga Boisserand[2], Luiz Henrique Medeiros Geraldo[3,4], Jose de Brito Neto[1,4], Thomas Mathivet[3], Salli Antila[5], Besma Barka[1], Yunling Xu[3], Jean-Mickael Thomas[6], Juliette Pestel[1], Marie-Stéphane Aigrot[1], Eric Song[7], Harri Nurmi[5], Seyoung Lee[2], Kari Alitalo[5], Nicolas Renier[1], Anne Eichmann[3,8] & Jean-Leon Thomas[1,2]*

Cranial lymphatic vessels (LVs) are involved in the transport of fluids, macromolecules and central nervous system (CNS) immune responses. Little information about spinal LVs is available, because these delicate structures are embedded within vertebral tissues and difficult to visualize using traditional histology. Here we show an extended vertebral column LV network using three-dimensional imaging of decalcified iDISCO+-clarified spine segments. Vertebral LVs connect to peripheral sensory and sympathetic ganglia and form metameric vertebral circuits connecting to lymph nodes and the thoracic duct. They drain the epidural space and the dura mater around the spinal cord and associate with leukocytes. Vertebral LVs remodel extensively after spinal cord injury and VEGF-C-induced vertebral lymphangiogenesis exacerbates the inflammatory responses, T cell infiltration and demyelination following focal spinal cord lesion. Therefore, vertebral LVs add to skull meningeal LVs as gatekeepers of CNS immunity and may be potential targets to improve the maintenance and repair of spinal tissues.

[1] Université Pierre et Marie Curie Paris 06 UMRS1127, Sorbonne Université, Institut du Cerveau et de la Moelle Epinière, Paris, France. [2] Department of Neurology, Yale University School of Medicine, New Haven, CT 06511, USA. [3] INSERM U970, Paris Cardiovascular Research Center, 56 Rue Leblanc, 75015 Paris, France. [4] Institute of Biomedical Sciences, Federal University of Rio de Janeiro, Rio de Janeiro, Brazil. [5] Wihuri Research Institute and Translational Cancer Medicine Program, Faculty of Medicine, University of Helsinki, Helsinki, Finland. [6] Ecole Nationale Supérieure d'Art de la Villa Arson, 06100 Nice, France. [7] Department of Immunology, Yale University School of Medicine, New Haven, CT 06510-3221, USA. [8] Cardiovascular Research Center and the Department of Cellular and Molecular Physiology, Yale University School of Medicine, New Haven, CT 06510-3221, USA. *email: jean-leon.thomas@yale.edu

The lymphatic vasculature controls fluid homeostasis, macromolecular clearance, and immune responses in peripheral tissues[1,2]. The brain was long considered to lack lymphatic vasculature, which has raised questions about how the cerebral interstitial fluid is cleared of waste products[3,4] and how immune surveillance of the brain is maintained[5–7]. This fluid is formed by water and small solutes that are exchanged through the capillary walls between the blood vessels and the brain. It has a similar composition to the cerebrospinal fluid (CSF) which drains the brain ventricles and meninges and is mainly produced in the choroid plexus[8]. The CSF has been proposed to dynamically exchange with interstitial fluid along glial lymphatic (glymphatic) non-vascular periarterial routes, without crossing the endothelial cell layer, and subsequently to be cleared from the brain into the subarachnoid space via similar perivenous routes[6,9]. The CSF outflow system involves specific extracranial lymphatic vasculature beds[10,11]. The recent identification of cranial meningeal LVs (mLVs) established another pathway for CSF outflow into deep cervical lymph nodes (dcLNs)[12–15]. In mice, cranial mLVs are mainly aligned alongside large dural venous sinuses, meningeal arteries and cranial nerves. Along the sagittal suture, the cranial lymphatic vasculature is valveless with small-diameter LVs, while it forms a larger network with valves and capillaries located adjacent to the subarachnoid space toward the basal aspects of the skull[12–16]. Cranial mLVs in the basal parts of the skull were initially shown to transport fluorescent tracers toward dcLNs via foramina at the base of the skull[12]. The basal mLVs include capillaries located adjacent to the subarachnoid space that have button-like junctions, allowing CSF uptake for the clearance of CSF macromolecules[15]. Using multiphoton microscopy, macromolecule and cell transport was reported also in mLVs alongside the superior sagittal and transverse sinuses[17], and consistent results were obtained by MRI imaging of primate and human mLs[16]. Meningeal lymphatic vasculature also exists in the skull of primates, including common marmoset monkeys and humans[14,16].

VEGF-C expression in vascular smooth muscle cells and VEGFR3 in lymphatic endothelial cells (LECs) are essential for the development of cranial mLVs[14,18]. The meningeal lymphatic vasculature develops later than the rest of the lymphatic network, first appearing at birth in the basal parts of the skull, then expanding during the neonatal period along dural blood vessels whose vascular smooth muscle cells supply the VEGF-C[14]. Immuno-histology on whole-mount preparations or cryosections showed that, during the first weeks after birth, LVs also developed a large network closely attached to the vertebral column[14]. These vertebral lymphatic vessels (vLVs) occur mainly in intervertebral spaces, having different morphology ventrally and dorsally, as well as along spinal nerve rami when exciting the spinal canal. Cranial mLVs located dorsally around the cisterna magna and ventrally around the foramen magnum appeared to be connected to vertebral LVs[14]. Further characterization confirmed that lymphatic vasculature extended caudally into into the whole vertebral canal and connected from there to the peripheral lymphatic vessels, as proposed by seminal papers[19,20]. We wanted to produce a three-dimensional (3D)-map of the vertebral lymphatic system that respects structural interactions between the spinal cord and meninges, the surrounding bone and mesenchymal environment and the neighboring peripheral nervous system (PNS). This required us to preserve the overall bone structures around the CNS while simultaneously accessing and labeling the LVs of meninges contained within the protective layers of muscular and skeletal tissues. To do so, we used the iDISCO+ technique, which enables volume imaging of immunolabeled structures in complex tissues[21,22]. Imaging of iDISCO+ treated vertebral segments with a light-sheet fluorescent microscope

(LSFM) revealed an extensive lymphatic vasculature inside the vertebral canal.

Here, we report that vertebral lymphatics are predominantly localized in the epidural space above the dura mater and drain tracers injected into the thoraco-lumbar spinal cord toward thoracic mediastinal lymph nodes. In addition, we show that VEGF-C-induced vertebral lymphangiogenesis exacerbates immune-cell infiltration and cytotoxic demyelination of spinal cord lesions in the lysolecithin (LPC)-induced focal demyelination model[21]. In the CNS, photoablation of the skull meningeal lymphatic vasculature has been reported to reduce the inflammatory response of brain-reactive T cells around demyelinated lesions in the EAE (experimental autoimmune encephalomyelitis) model of multiple sclerosis[22]. Therefore, the vertebral lymphatic system conveys an additional remote control of immune surveillance to the CNS.

## Results

**Lymphatic vasculature pattern in the thoracic spine.** To label vascular, immune and neural cell compartments within the intact vertebral column, segments of 2–4 vertebrae were dissected together with the surrounding muscle tissue and decalcified in Morse's solution[23]. iDISCO+ tissue clearing and immunolabeling followed by light-sheet fluorescence microscope (LSFM) imaging were then used for 3D-reconstruction of the spinal LV network.

The iDISCO+ protocol was first applied to the thoracic spine, with the goal to characterize the 3D anatomy of vLVs. Figure 1a, b illustrates a lateral view of Alizarin red staining of bones within a cleared spinal column segment to reveal the vertebrae, intervertebral spaces and ligamentum flavum. Figure 1c shows a schematic latero-frontal perspective view of a thoracic vertebral segment. Lymphatic endothelial cells (LECs) were labeled using polyclonal antibodies against two well-established LEC markers, the LYVE1 cell surface receptor and the nuclear PROX1 transcription factor[24–26]. PROX1-labeled LYVE1-positive LECs and LYVE1-negative cells within the spinal cord that were previously identified as oligodendroglial cells[27] (Supplementary Fig. 1a, b). LYVE1 labeled PROX1-positive LECs and PROX1-negative myeloid cells, as previously reported[27] (Supplementary Fig. 1a, b). LYVE1-positive LECs within the vertebral column were negative for the blood vessel marker Podocalyxin[28] in the present conditions of immunolabeling (Supplementary Fig. 1c–h).

Despite labeling of some non-LECs, both markers clearly revealed a dense lymphatic network that was present between vertebrae and appeared mainly confined to the intervertebral spaces (Fig. 1d, e and Supplementary Movies 1, 2)[14]. A few longitudinal vessels linked adjacent intervertebral lymphatic circuits together along the spinal cord (salmon arrows in Fig. 1e, f). Vertebral LVs (vLVs) were also connected to the peripheral lymphatic system surrounding the vertebrae, dorsally through the ligamentum flavum (Fig. 1f), dorsolaterally along the dorsal facet joint and ventrolaterally through the intervertebral foramen along ventral nerve rami (Fig. 1e, f).

We next used Imaris-3D software to illustrate the anatomy of vLV circuits. We used global image acquisitions of the thoracic spine (see Supplementary Movie 2), showing a succession of vertebral lymphatic units along the rostro-caudal axis. Images were then segmented to generate a color-coded map of vLV circuits. In Fig. 1g, h and Supplementary Movies 3, 4, each color defines the PROX1+ pattern of one vertebra along three successive thoracic vertebrae (red, blue, green) as well as the peripheral lymphatic vasculature (white). This confirms a metameric organization of vLVs[14].

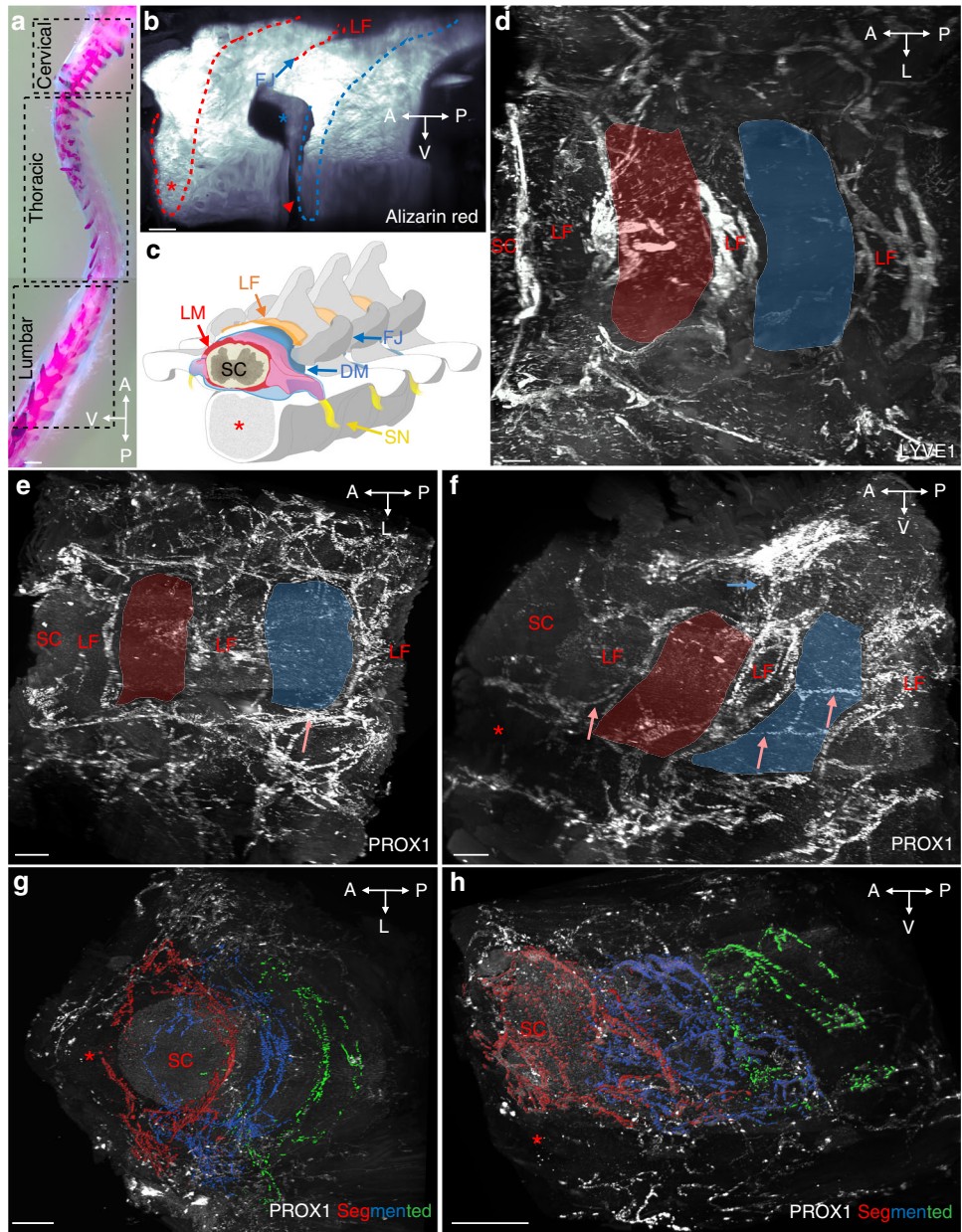

**Fig. 1** Segmental pattern of the vertebral lymphatic vasculature in the thoracic spine. **a** Alcian blue/Alizarin red staining of the mouse vertebral column with boxes indicating position of images shown in Figs. 1–4 (thoracic vertebrae) and 5 (cervical and lumbar vertebrae), spatial orientation (A: anterior, P: posterior, L: lateral, V: ventral). **b** Alizarin red staining of two successive thoracic vertebrae (delimited by red/blue dots, lateral view). LF: ligamentum flavum, red asterisk: ventral vertebral body, blue arrow: facet joint (FJ), red arrowhead: ventral intervertebral disk, blue asterisk: intervertebral foramen. **c** latero-frontal schematic drawing corresponding to (**b**). DM: dura mater, LM: leptomeninges (pia mater and arachnoid), SC: spinal cord, SN: spinal nerve. **d** Dorsal view of LYVE1 staining. Red and blue areas correspond to two successive vertebrae. Note LVs lining ligamentum flavum. **e**, **f** Dorsal (**e**) and lateral (**f**) views of the PROX1 expression pattern. Red and blue areas correspond to two successive vertebrae. Salmon arrows: intervertebral LVs, blue arrow: dorsal LVs. **g**, **h** Segmented images of the PROX1 LV network (fronto-dorsal (**g**) and lateral (**h**) views) highlighting three successive vertebral LV units (red, blue, green). Scale bars: 2 mm (**a**); 300 μm (**b**, **e**, **f**); 200 μm (**d**, **g**, **h**)

**Modular architecture of vertebral lymphatic vasculature.** We next mapped the vLV network in the vertebral canal from the dorsal to the ventral part of a vertebra. Supplementary Movie 2 and the corresponding view in Fig. 2a, b show vLVs around one segment of the thoracic spinal cord, and areas where higher magnification views were taken. Dorsally, semicircular lymphatic vessels navigate around the spinal cord (Fig. 2c). At the ventral border of the ligamentum flavum, located at the dorsal midline between two spinous processes, these vLVs contact lymphatic branches entering the epidural space from the overlying dense

peripheral lymphatic vasculature (Fig. 2d). Laterally, at the level of the transverse facet joints, semicircular vessels including peripheral lymphatic vessels from the dorsal plexus converge toward a lymphatic circle (blue arrow in Fig. 2c). From this point, vLVs distribute either radially toward the periphery, or ventrally toward the emergence of the dorsal spinal nerve roots (red double arrow in Fig. 2c). Supplementary Movie 2 allows to follow peripheral lymphatic vessels from the dorsal midline of intervertebral ligaments to the lateral exit points of the vertebral canal. At the intervertebral foramen, DRGs are covered by vLVs that converge

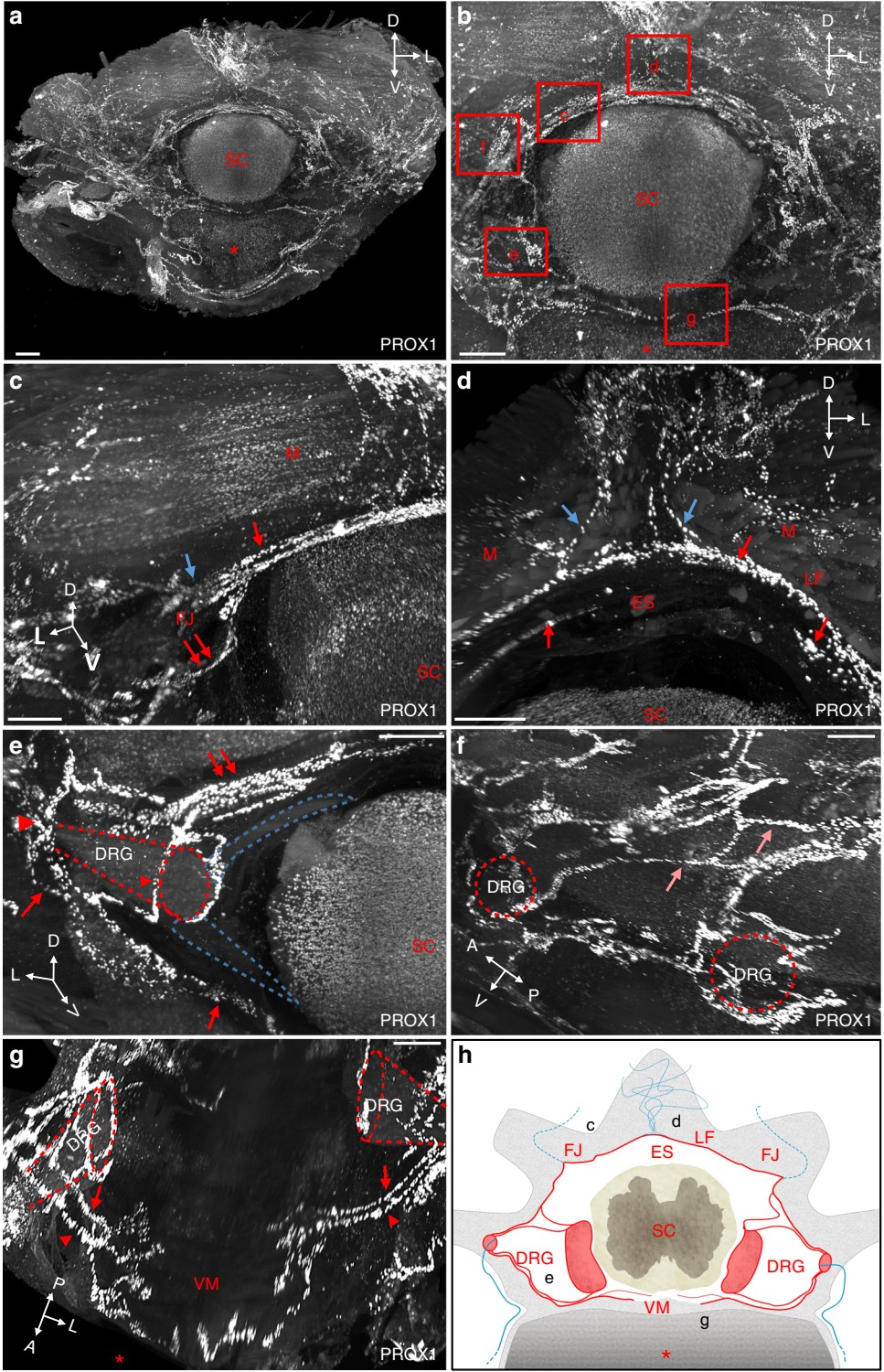

**Fig. 2** Modular architecture of vertebral lymphatic vasculature. **a** Frontal view of a cleared thoracic vertebra stained with an anti-PROX1 antibody. Red asterisk: vertebral ventral body, SC: spinal cord spatial orientation (D: dorsal, L: lateral, V: ventral). **b** Magnifications of red boxed areas are shown in (**c–g**). **c** Semicircular dorsal LVs (red arrow) surround the spinal cord, exit dorsolaterally (blue arrow) and also extend a latero-ventral connection to the dorsal nerve root (double red arrows in (**c**) and (**e**)). Note PROX1+ cells in SC and perivertebral muscles (M), FJ: facet joint. **d** At the ventral face of the ligamentum flavum (LF) located between two spinous processes, dorsal LVs (blue arrows) enter the vertebral canal and join semicircular LVs (red arrows). Note circles of LVs bordering the upper side of the epidural space (ES). **e** Ventrolateral LV circuitry around DRG (red arrowheads). Blue dotted-lines: spinal nerve roots, red dotted-lines: DRG. **f** Lateral view with intervertebral LVs (salmon arrows). **g** Two ventral branches (red arrows and arrowheads) run between each side of the ventral midline (VM) and the DRG. **h** Schematic representation of a frontal view of a thoracic vertebral LV unit. Longitudinal connecting vessels between vertebral units are not represented. FJ: facet joint; LF: ligamentum flavum; ES: epidural space; DRG: dorsal root ganglia; VM: ventral midline; SC: spinal cord. Black letters refer to images in (**c–g**). Scale bars: 300 μm (**a–g**)

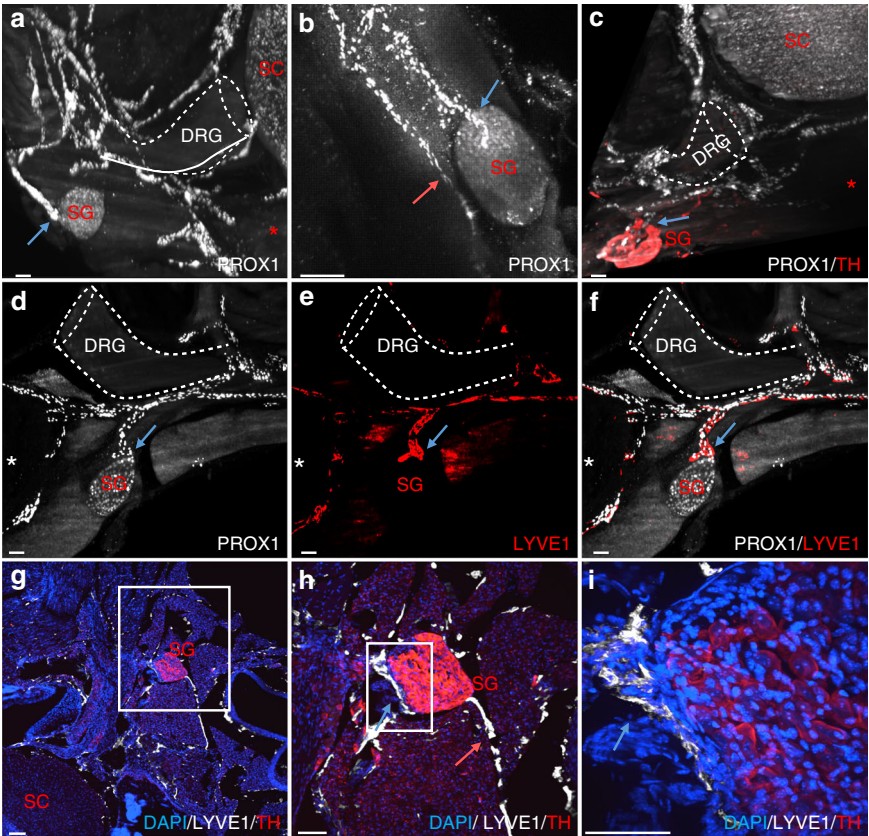

**Fig. 3** Vertebral lymphatic vessel connections with sympathetic ganglia. **a**, **b** Thoracic PROX1+ LVs contact paravertebral sympathetic ganglia (SG) (blue arrows). White dotted-lines: DRG. **c** PROX1 (white) and tyrosine hydroxylase (TH, red) double labeling shows a ventral LV branch contacting a paravertebral TH+ sympathetic ganglion (blue arrow). **d**–**f** PROX1 (white)/LYVE1 (red) double labeling of the LV-SG connection (blue arrow). **g**–**i** 2D-confocal images of cervical cryosections labeled with LYVE1 (white), TH (red), and DAPI (blue). White box: area magnified in (**h**) and (**i**). A LV contacts a TH+ SG (blue arrow). Note a second ventral LV branch running along the SG, without entering its cortical layer (**h**, salmon arrow). This branch is also seen in panel (**b**) (salmon arrow). White dotted-lines: DRG; Red asterisk: vertebral ventral body; SC: spinal cord. Scale bars: 100 μm (**a**–**h**); 200 μm (**i**)

from the ventral and dorsolateral circuits at their proximal and distal end, respectively (arrowheads in Fig. 2e). In addition to these two circuits, a few longitudinal connecting vessels link vertebral lymphatic units together (Fig. 2f). Ventrally, a second circuit of semicircular lymphatic vessels converges toward the ventral spinal nerve rami exit, while no lymphatic vessels are observed at the ventral midline (Fig. 2g)[14]. We observed a similar vLV pattern in other thoracic segments ($n = 5$) allowing us to generate a schematic representation of the different compartments of the vLV circuitry with a specific color code for the intervertebral circuits (red) and vertebral branches of the peripheral LV (blue) (Fig. 2h).

To verify that the iDISCO+ protocol preserved the integrity of tissue structure and anatomy[29], we performed additional immunostainings on decalcified EDTA-treated vertebral segments that were either cryoprotected and frozen, or dehydrated and embedded in paraffin. Confocal and conventional microscopy imaging of sections perfectly reproduced the 3D-images collected with a LSFM on IDISCO+ treated samples (Supplementary Fig. 2) and thus substantiated the iDISCO+ based-model described above.

**Vertebral lymphatic vessels contact sympathetic ganglia.** vLVs covering the dura mater of DRGs (Fig. 2e) extend collateral branches bilaterally along the spine, which contact paravertebral PROX1low ganglia (Fig. 3a, b). PROX1 is known to be expressed in the sympathetic neuronal lineage[30] and double labeling with antibodies against PROX1 and tyrosine hydroxylase (TH), a specific marker of adrenergic nerves and ganglia, confirmed that specific branches emerging from vLVs connected to TH+/PROX1low sympathetic ganglia (Fig. 3c). On each side of the spinal cord, PROX1+/LYVE1+ LVs contacted one sympathetic ganglion per spinal level (Fig. 3d–f). Complementary analyses by high resolution confocal imaging on vertebral column cryosections showed that the connection between LVs and sympathetic ganglia (Fig. 3g, h) occured at the surface of the ganglion cortical layer (Fig. 3i). These data reveal a hitherto unknown anatomical interaction between the autonomous nervous system and lymphatic vessels derived from vLVs.

**vLV patterning differs between vertebral column levels.** We next examined the lymphatic vasculature at the cervical and thoracic level of the vertebral column. Stereomicroscopic imaging of whole-mount preparations revealed LVs around the cisterna magna and within the vertebral canal, where they located at the level of intervertebral ligaments and surrounding spinal nerve rami (Supplementary Fig. 3a–d). The LV patterning in cervical vertebrae and along the vertebral column was then analyzed in further detail by volume imaging.

In the cervical region, we observed a dorsal extravertebral lymphatic plexus (blue arrow, Fig. 4a) as well as intravertebral vLVs that exited ventrally and bilaterally (red arrows) through the intervertebral foramen to ventrally (salmon arrow) connect to dcLNs (green in Fig. 4b; Supplementary Fig. 3e, f). Thoracic vLVs were defined by a large dorsal extravertebral plexus (blue arrow, Fig. 4c) and a direct connection from ventrolateral DRG LVs (red

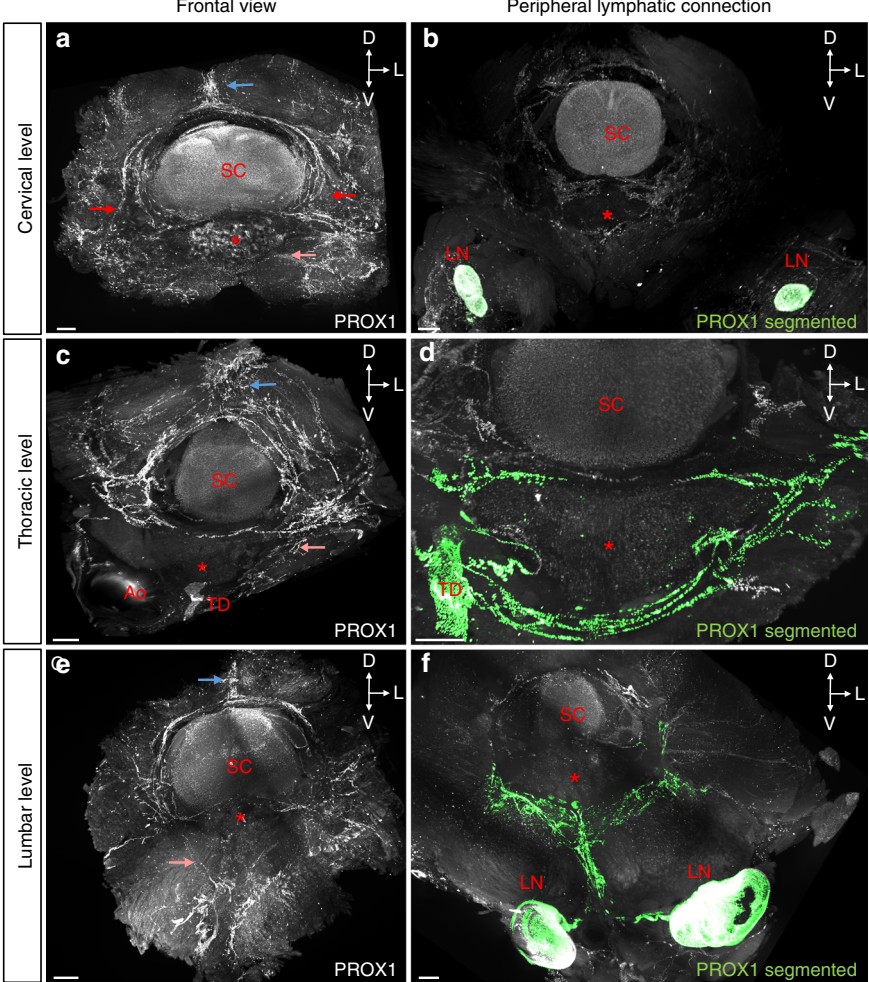

**Fig. 4** Variations of vLV pattern along the vertebral column. **a–f** Pattern of PROX1⁺ LVs in the cervical (**a**, **b**), thoracic (**c**, **d**), and lumbar (**e**, **f**) vertebral column, spatial orientation (D: dorsal, L: lateral, V: ventral). Left panels show frontal views, right panels show connection to peripheral lymph nodes (LN) and thoracic duct (TD). **a**, **c**, **e** Note fewer LVs in the dorsal plexus between intervertebral spinous processes of cervical and lumbar vertebrae compared to thoracic ones (blue arrows). **a** LVs exit bilaterally (red arrows) through the intervertebral foramen. **a**, **c**, **e** Also note differences in ventral root exit circuits between regions (salmon arrows). **b**, **d**, **f** LV ventral exit circuits (green) to **b** deep cervical LNs, **d** thoracic duct, or **f** to renal LNs. Red asterisk: vertebral ventral body; SC: spinal cord; Ao: Aorta. Scale bars: 300 μm (**a–f**)

arrow) to the thoracic lymphatic duct (green in Fig. 4d). The thoracic and lumbar regions displayed similar extensions and patterns of extravertebral and intravertebral LVs (Fig. 4c, e). In lumbar vertebrae, the ventrolateral circuits that exited on each side of the vertebral canal connected to lymph nodes. As shown in green in Fig. 4f, lymphatic vessels circumvented the ventral body of the lumbar vertebra (salmon arrow), converged on the ventral midline and split into two branches running toward the pair of renal lymph nodes. Therefore, the vLV architecture is conserved along the vertebral column, but the extension of extravertebral and intravertebral vessels around the spinal cord and their connection to the peripheral lymphatic system differs between the cervical, thoracic and lumbar vertebral levels.

**Epidural and dural vertebral lymphatic vessels**. To obtain a 3D annotation of vLV localization in the spinal canal and meninges, PROX1-labeled vertebral volumetric images were used to generate segmented images of membranes and the epidural space around the spinal cord. We manually annotated in 3D the meninges, the epidural space and ligamentum flavum (Fig. 5a). Figure 5a shows one image slice with a color code for meningeal layers (purple area) and the epidural space (green area). We also

present color-coded layer masks for the arachnoid and dura mater together (purple area in Fig. 5b), or the dura mater and the epidural space together (green area in Fig. 5c). The overlay of both masks revealed that PROX1⁺ vLVs mainly localized in the epidural space (green), while the underlying dura mater layer (gray) includes ventral vLVs (white) around DRGs (Fig. 5d and Supplementary Movie 5). As shown on a lateral view (Fig. 5e), dura mater vLVs localized most extensively at bilateral DRGs. Interestingly, connecting vessels between two successive vertebrae (salmon arrows in Fig. 5e) navigated in the epidural space and appeared to join vLVs of the dura mater close to the DRGs (white arrows in Fig. 5e), suggesting a possible confluence of peripheral lymph and CSF at this level. Scheme representing the anatomy of vLVs in the thoracic vertebral canal is shown in Fig. 5f.

Complementary examination of whole-mounted spinal cord meninges (Supplementary Fig. 4a) and cryo/paraffin sections (Supplementary Fig. 4b–g) confirmed that the dura mater lymphatic vessels were restricted around the DRGs and spinal nerve rami and located on the dorsal surface of the dura mater (Supplementary Fig. 4c, f), which is not in direct contact with the CSF.

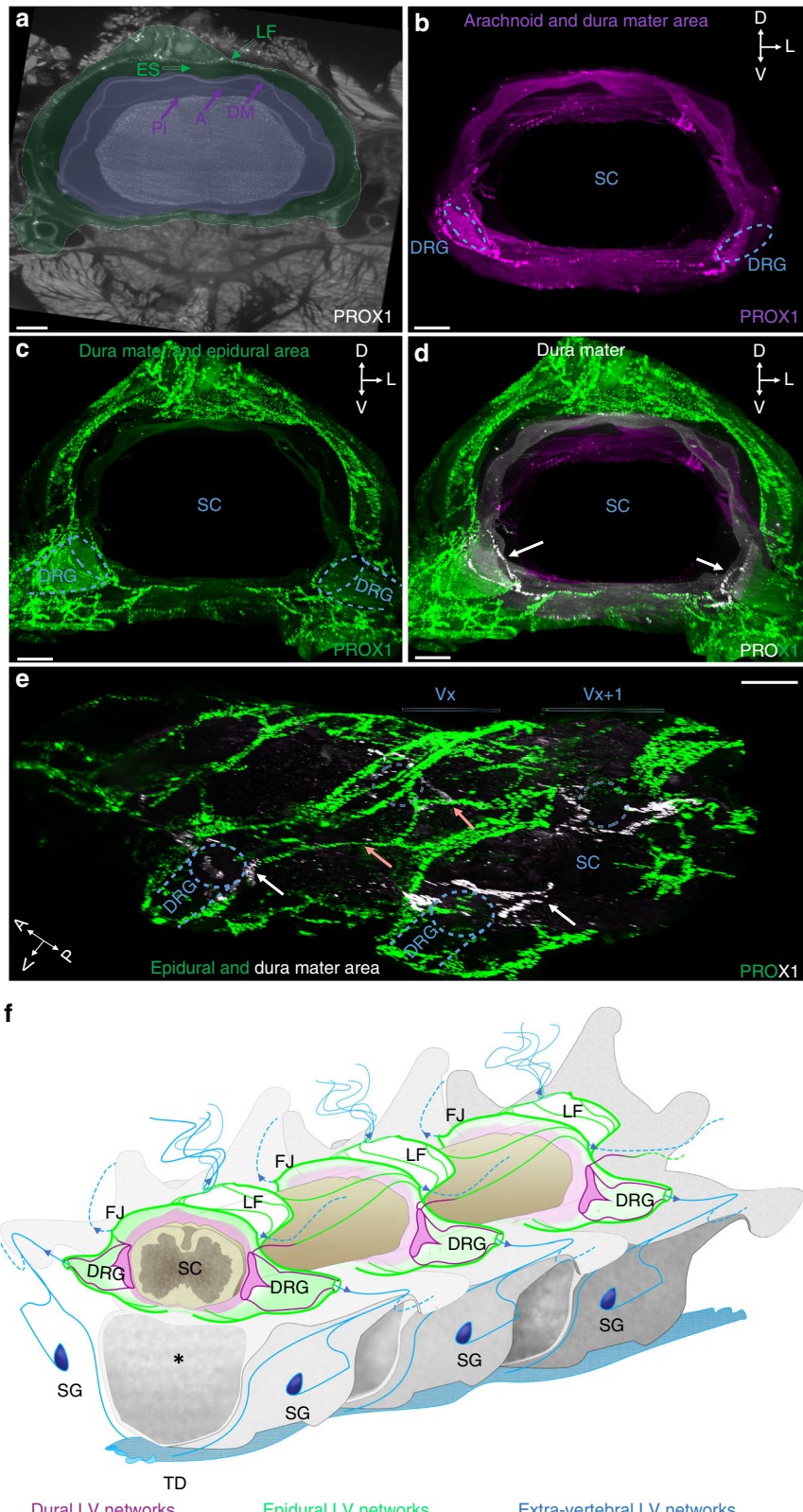

## vLV-mediated drainage of the vertebral column.

The function of the vLV system was explored by testing the drainage potential of epidural and dura mater vLVs. Molecular tracers were injected into one side of the spinal cord parenchyme at the thoraco-lumbar level, and their distribution around the injection site was examined 15 and 45 min after injection (a.i.) (Fig. 6a). We used as molecular tracers either LYVE1 antibodies that were detected with a secondary antibody, or fluorescent albumin (OVA-A[555]). It is worth noting that this surgery procedure punctures the dura mater, which allows access of injected tracer into the epidural space located above spinal meninges, locally at the puncture site. LSFM imaging of iDISCO +-treated vertebral samples revealed

**Fig. 5** Epidural and dural lymphatic circuits of the spine. **a** 2D-single frontal image slice (2 μm thick) of the cervical vertebral column with enhanced brightness to reveal PROX1-expressing nuclei and spinal cord (SC), meninges including pia mater (Pi), arachnoid (A) and dura mater (DM), the epidural space (ES), and the ligamentum flavum (LF). A color-coded segmentation of layers around the spinal cord shows the meningeal layers in purple and the dura mater plus the epidural space in green. **b–d** 3D-reconstruction of frontal images of the cervical vertebral column with color-coded layers: the arachnoid and dura mater in purple (**b**); the dura mater and epidural space in green (**c**); combined layer marks showing the arachnoid in purple, the dura mater in white, and the epidural space in green (**d**), spatial orientation (D: dorsal, L: lateral, V: ventral). A noticeable LV network fills the epidural space (green) while dura mater LVs (white) are mainly restricted to DRGs (white arrows) and few branches on each side of the dorsal and ventral midline. **e** 3D-reconstruction of lateral images of the thoracic vertebral column with color-coded layers illustrated in (**d**). Blue dotted-lines: bilateral DRGs; salmon arrows: intervertebral LVs; Red asterisk: vertebral ventral body. Vx: vertebra x, Vx + 1: vertebra x + 1. **f** Schematic representation of the lymphatic vasculature in the thoracic vertebral column. LVs are present in the epidural space (green) around the spinal cord and in the dura mater (purple). Extravertebral LVs extend dorsal processes (blue) and ventral connections with sympathetic ganglia (SG, deep blue) and the thoracic duct (TD, light blue). Blue arrowheads; exit points of vertebral lymphatic circuits; Blue dots: connections with extravertebral lymphatic networks; Black asterisk: vertebral ventral body; DRG: dorsal root ganglia; FJ: facet joint; LF: ligamentum flavum; SC: spinal cord; SG: sympathetic ganglia. Scale bars: 300 μm (**a–e**)

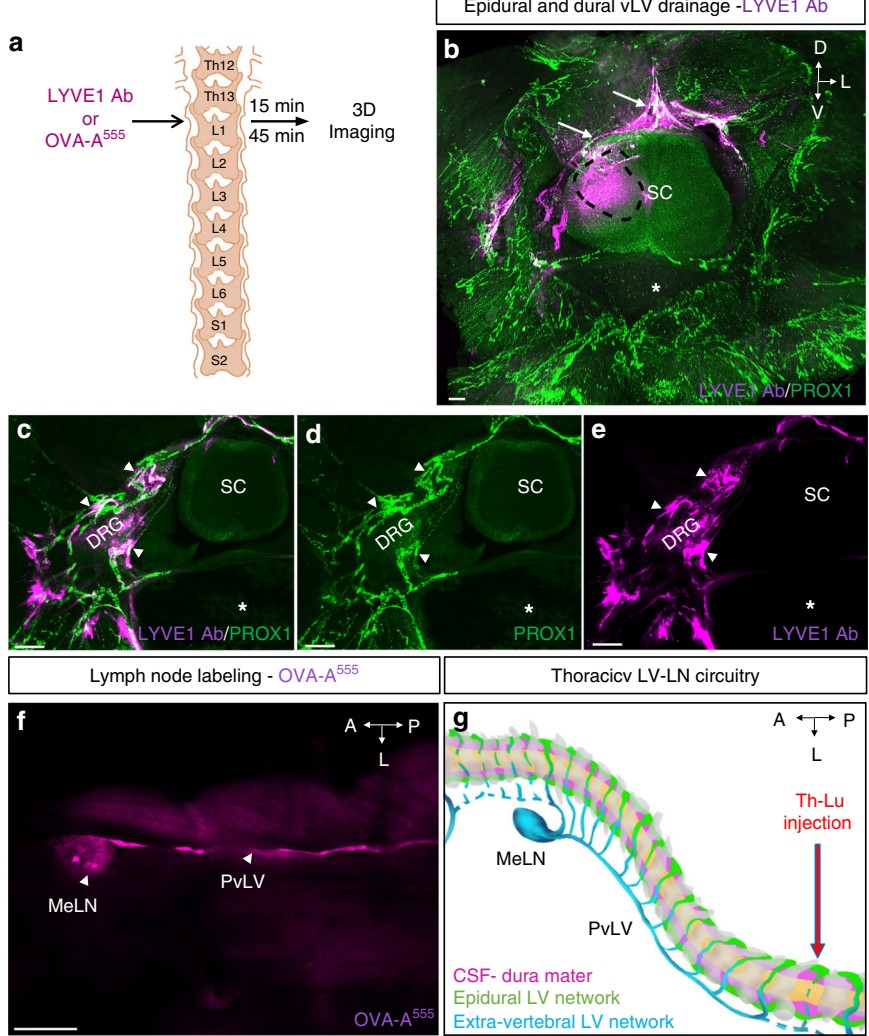

**Fig. 6** Epidural and dural lymphatic drainage. **a** Schematic representation of the procedure used to test spinal cord drainage: LYVE1 Ab or OVA-A$^{555}$ was injected in the thoraco-lumbar (Th-Lb) region of the spinal cord parenchyme. Fifteen or forty-five minutes later, mice were sacrificed. LYVE1 was detected by labeling with a secondary antibody, while OVA-A$^{555}$ was directly identified by fluorescence detection. Schema is adapted from Fig. 2c in ref. [59]. **b–e** LYVE1 Ab uptake by epidural and dural lymphatic circuits after 45 min. **b** LYVE1 Ab (purple) injection area (black dotted line) and epidural LVs recapture (white arrows). **c–e** Colocalization of LYVE1 with LVs around DRG, in contact with dura mater (white arrowheads). Asterisk: vertebral ventral body, SC: spinal cord. **f** OVA-A$^{555}$ injection leads to labeling of ipsilateral mediastinal lymph node (MeLNs) after 15 min. **g** Schematic representation of the lymphatic drainage pattern in the thoracic vertebral column. Thoracic vertebrae (gray), dura mater and CSF (purple), LVs of epidural space (green), extravertebral LVs and lymph node (blue). Blue dots: accumulation of OVA-A$^{555}$ tracer in LVs and lymph node. MeLN: mediastinal lymph node; PvLV: paravertebral lymphatic vessel. Scale bars: 200 μm (**b–e**); 1 mm (**f**)

that the injected markers localized in and around PROX1[+] vLVs of the epidural space and dura mater (white arrows, Fig. 6b–e and Supplementary Movie 6). Confocal imaging of decalcified and frozen samples showed that OVA-A[555] localized within the vLV lumen (Supplementary Fig. 5), thus demonstrating the uptake and drainage properties of vLVs.

15 min after OVA-A[555] injections into the thoraco-lumbar spinal cord, tracer accumulated in the ipsilateral paravertebral lymphatic vessel and mediastinal lymph node, in 9 out of 12 cases (Fig. 6f). Therefore, vLVs provide a regional outflow for epidural fluids toward lymph nodes. A schematic model of the thoraco-lumbar lymphatic drainage circuitry is shown in Fig. 6g.

**Vertebral LVs respond to VEGF-C and spinal cord injury.** To assess the dependence of vLVs on VEGF-C, we generated gain-of-VEGF-C signaling mice by either intra-cisterna magna (i.c.m.) or lumbo-sacral (l.s.) injection of adeno-associated viral vectors (AAVs) encoding mVEGF-C (AAV-mVEGF-C)[15] (Fig. 7a, d). Control mice were injected with AAVs encoding soluble mVEGFR3_{4−7}-Ig (VEGFR3 ectodomains that do not bind VEGF-C) (AAV-control)[31]. One month later, mice were analyzed by PROX1 immunostaining. Compared to controls, VEGF-C injected mice showed a strongly expanded vLV network, in particular of dorsolateral lymphatic rings in the intervertebral disk of cervical vertebrae after i.c.m. injection (Fig. 7b, c and Supplementary Movie 7), and lumbar vertebrae after l.s. injection (Fig. 7e, f).

To determine if adult vLVs might respond to spinal cord injury, we injected LPC (1 μl) into one side of the spinal cord at the thoraco-lumbar level (Fig. 7g). LPC is toxic to oligodendrocytes and rapidly induces demyelinating spinal cord lesions (Fig. 7i inset)[32,33]. Within a week after the surgery, a robust extravertebral and intravertebral growth in LVs was induced in LPC-injured mice (Fig. 7h–i and Supplementary Movie 8). To quantify the response, we opened the vertebral column to expose intravertebral vLVs that were stained with LYVE1 on whole-mount preparations, followed by vessels diameter and surface area measurements (as red stippled area in Fig. 7i). Pairwise Mann–Whitney U test comparison to control mice revealed a significant increase in vLV area after LPC injury ($p = 0.0286$), however, this did not reach statistical significance in a mutliple group comparison (Supplementary Fig. 6e). LPC-lymphatic vessel diameter and area were not affected by control AAV-mVEGFR3_{4−7}-Ig (LPC^{control}), while they were significantly enhanced in mice pretreated with AAV-mVEGF-C (LPC^{VEGF-C} mice) for one month (Fig. 7j, Supplementary Fig. 6e). LPC injury in mice pretreated by lumbo-sacral (l.s.) injection of AAVs encoding soluble mVEGFR3_{1−3}-Ig (LPC^{VEGF-C trap} mice) for one month resulted in reduction of vLV diameter that was significantly different from LPC^{control} mice (Fig. 7j). LPC^{VEGF-C trap} mice showed a reduction of vLV area that was significantly different from LPC^{control} mice in a pairwise Mann–Whitney U-test comparison ($p = 0.0286$), but failed to reach statistical significance in a multiple group comparison (Supplementary Fig. 6e). However, LPC injury in K14-VEGFR3-Ig homozygous mice, in which endogenous VEGF-C/VEGF-D ligands are constitutively trapped to prevent VEGFR3 signaling[31], was associated with a significant reduction of vLV diameter and area compared to heterozygous littermates (Fig. 7k, Supplementary Fig. 6f). Taken together, these data show that vLVs respond to VEGF-C and spinal cord injury.

**Effects of vLVs on myeloid and lymphoid cells.** We next asked whether vLVs concentrated myeloid and lymphoid cells, as previously reported for skull lymphatics[13]. Pre-cleared segments of the vertebral column co-labeled with antibodies against LYVE1

and the common leukocyte antigen CD45 showed that CD45[+] leukocytes inside the vertebral canal were concentrated around LYVE1[+] vLVs (Fig. 8a–d). On cryosections we observed leukocytes concentrated close to, or within, *Vegfr3:YFP*[+] lymphatic vessels in intervertebral ligaments (Fig. 8e). CD45[+] leukocytes included around 40% of C11b[+] macrophages, 40% of CD3[+] T cells, and 20% of CD19[+] B cells. Furthermore, around 40% of CD45[+] leukocytes expressed MHCII (Supplementary Fig. 6a–d).

Leukocyte numbers and ratios were similar between control, AAV-mVEGF-C and K14-VEGFR3-Ig homozygous and heterozygous mice (Supplementary Fig. 6a–d). In contrast, LPC injury induced a strong increase in leukocyte numbers around vLVs that was further enhanced by AAV-mVEGF-C and reduced by mVEGF-C trap (Fig. 8f).

The size of demyelinated lesions, identified as spinal white matter areas devoid of MBP (Myelin Basic Protein) expression, was significantly increased in LPC^{VEGF-C} mice compared to control AAV treated mice (Fig. 8g, h). LPC^{VEGF-C trap} mice mice showed a significantly reduced lesion size when compared to LPC^{VEGF-C} mice, and a slight but not significant reduction in lesion area when compared to LPC^{control} mice (Fig. 8g, h).

Further quantifications were done on spinal cord sections in the lesioned area vs. the contralateral uninjured side. As expected, LPC injection reduced the number of NeuN-positive neurons in the peri-lesional area compared to the uninjured side (Supplementary Fig. 7a). Pre-treatment with AAV-control or with AAV-mVEGF-C trap had no effect compared to LPC injury alone, while AAV-mVEGF-C further reduced the number of NeuN-positive neurons on the injured side, indicating deleterious effects of expanded vLVs on spinal cord demyelinating lesions (Supplementary Fig. 7a). LPC injection induced an increase in the number of F4/80[+] microglia and monocyte-derived macrophages, Iba1[+] microglia, and CD3[+] T cells in the injected side compared with the contralateral side (Supplementary Fig. 7b–d). AAV-control had no effect on immune-cell numbers, while the numbers of F4/80[+], Iba1[+], and CD3[+] cells were further amplified in LPC^{VEGF-C} mice (Supplementary Fig. 7b–d). LPC^{VEGF-C trap} mice showed significantly reduced leukocyte numbers when compared to LPC^{VEGF-C} mice, however, leukocyte infiltration was not reduced when compared to untreated LPC^{control} mice (Supplementary Fig. 7b–d). Infiltration of F4/80[+] macrophages/microglia and CD3[+] T cells was also not significantly different between K14 homozygous and heterzozygous LPC-injured mice (Supplementary Fig. 7e, f). Altogether, these results demonstrate that a gain-of-vLVs amplified the cytotoxic effect resulting from LPC-induced injury.

## Discussion

We here report the 3D anatomy and the function of vLVs in the vertebral canal (Fig. 5f). The data reveal an extensive and complex lymphatic vasculature in the vertebral column, surprisingly dense in comparison to the one that covers the cranial dura mater[15]. Previous literature reported the presence of LVs on whole-mount preparations of vertebral dura mater in monkeys[34] and on sections of intact and injured vertebral tissue in humans[35]. Pioneer studies in the late 19th and early 20th centuries[36–38] and later works of Ivanow[19] and Brierley and Fields[20] had investigated the flow of the lymph stream along the spine and in the spinal roots of the cord, providing evidence that lymphatics contribute to the propagation of infectious agents (toxins, polyomyelitis, tetanus)[39]. More recently, a continuum of metameric spinal lymphatics was described in the cervical, thoracic and lumbar areas of the vertebral column, both on dorsal and ventral sides, with lateral exits of the vertebral canal along blood vessels and spinal nerves[14]. We here extend these seminal findings by 3D-views of lymphatic

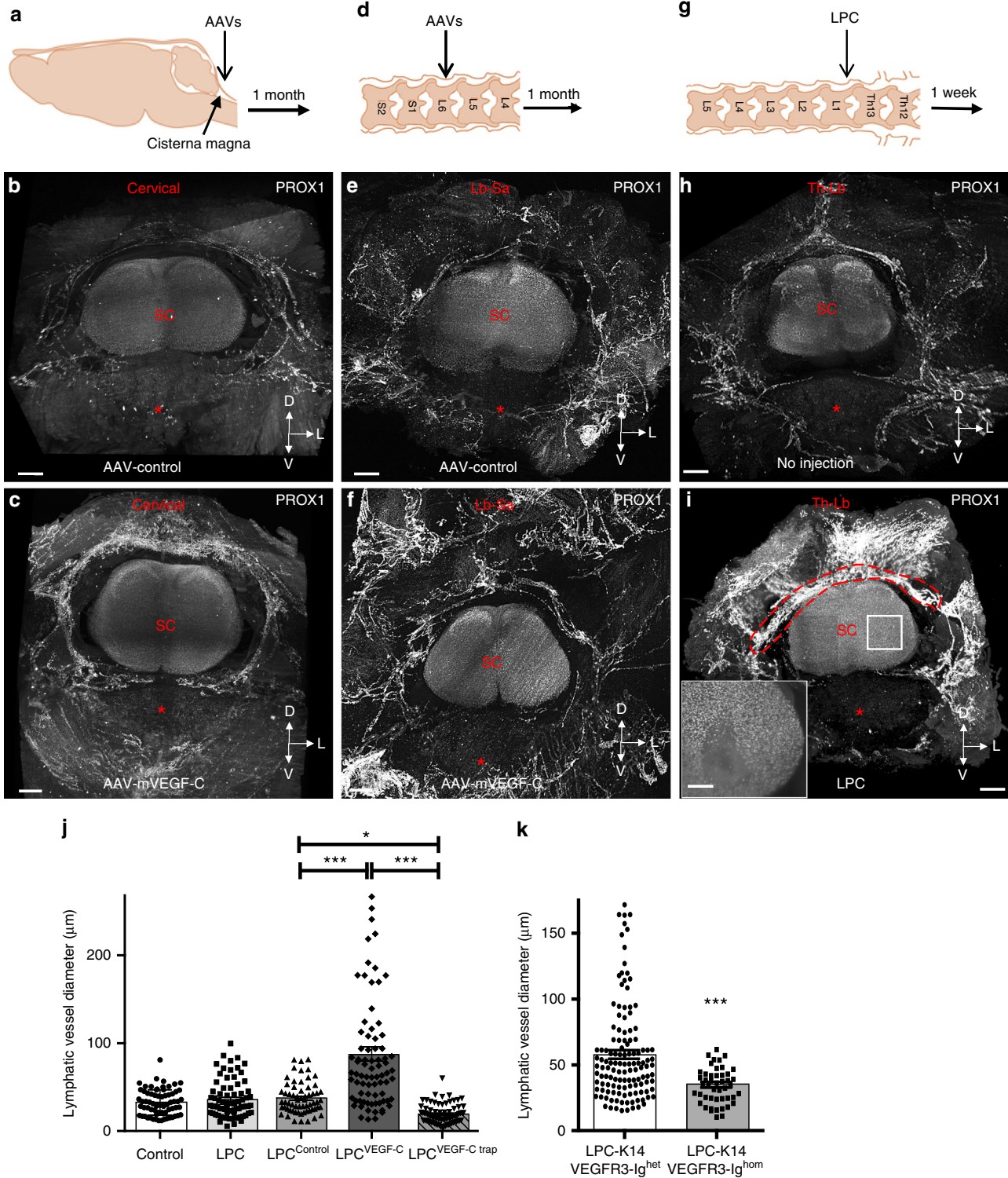

vasculature organization and function in vertebral canal drainage (Fig. 6g).

Each vertebra is drained by semicircular dorsal and ventral vessels, which exit the vertebral column at intervertebral foramina. vLVs extend along spinal nerve rami to reach lymph nodes adjacent to the cervical, thoracic and lumbar regions of the spine, as well as the thoracic lymphatic duct in the thoracic region. The vertebral lymphatic network is thus organized as a metameric network of peripheral LV-connected vertebral lymphatic units that are interconnected by a few thin longitudinal

vessels. The absence of large longitudinal dorsal or ventral LVs suggests that vLVs do not drain vertebral lymph as a continuous stream along the vertebral column axis, but rather at the level of each vertebra. This model is supported by our findings that vertebral lymphatic vasculature consists mainly of an extensive network of epidural vessels, located in the intervertebral tissue and beneath the ligamentum flavum, and which drain the epidural lymph of the vertebral column. These observations do not exclude that the spinal CSF may also follow a directional downward flux within the central canal and the spinal subarachnoid

**Fig. 7** vLVs are VEGF-C dependent and remodel after spinal injury. **a–f** VEGF-C induces epidural and dural lymphangiogenesis. **a–c** Cervical spine lymphangiogenesis after i.c.m. injection of AAV-mVEGF-C. **a** Schematic of i.c.m. injection to deliver AAVs into the CSF and toward the cervical spine. **b, c** LSFM coronal view of the cervical spine one month after AAV injection. Pattern of PROX1+ LVs (white) in AAV-control (**b**) and AAV-mVEGF-C (**c**) mice. Note that VEGF-C induced a robust epidural and dural lymphangiogenesis. **d–f** Thoraco-lumbar lymphangiogenesis induced by lumbo-sacral delivery of AAV-mVEGF-C. **d** Schematic of AAV injection sites into the lumbo-sacral spinal cord, adapted from Fig. 2c in ref. [59]. **e, f** Pattern of PROX1+ LVs (white) in AAV-control (**e**) and AAV-mVEGF-C (**f**) mice. White asterisk: vertebral ventral body; SC: Spinal cord. **g–i** Focal injury in the thoraco-lumbar spinal cord. **g** Schematic of LPC injection into the thoraco-lumbar spinal cord, adapted from Fig. 2c in ref. [59]. **h, i** Pattern of PROX1+ LVs (white) in control-non lesioned (**h**) and LPC-injected mice (**i**). Inset in (**i**) shows the spinal cord lesion (stippled area), spatial orientation (D: dorsal, L: lateral, V: ventral). **j, k** Quantification of lymphatic vessel diameter (red stippled area in (**i**)) after LPC-spinal cord injury in gain- and loss-of-mVEGF-C signaling mice (**j**) and in LPC-injured K14-VEGFR3-Ig^hom mice and -K14-VEGFR3-Ig^het (control) mice (**k**). $n = 4$ biologically independent mice/independent experiment, and data show mean+/−SD (error bar) in (**j, k**); one-way ANOVA with Tukey's multiple-comparisons test (**j**) and Mann–Whitney U test (**k**); *$p < 0.05$, ***$p < 0.001$. Source data are provided as a Source Data file. Scale bars: 300 μm (**b, c, e, f, h, i**).

space, toward the caudal end of the spine, as recently demonstrated by dynamic in vivo imaging of CSF flow after intraventricular CSF tracer injection[40]. In line with our observations, the authors report the outflow of CSF tracers from the spinal subarachnoid space from intravertebral regions of the sacral spine toward sacral and iliac LNs.

Interestingly, consistently with our earlier previous findings[14], we found vLVs in contact with dura mater (Supplementary Fig. 4c, f), where molecular tracers were also detected within vLVs, around DRGs and spinal nerve rami (Fig. 6b–e). Subsequent surgeries specifically delivering fluorescent tracers within the spinal cord subarachnoid space will be required to determine whether these vertebral dural vLVs may be hotspots contributing to CSF uptake toward lumbo-sacral LNs.

We note that there is a regional variation in the LV size, which is inversely correlated to the volume of CSF, with large cerebral ventricular volumes associated with a discrete network of cranial mLVs and a small vertebral ependymal volume correlated with a large vertebral lymphatic vasculature. One possible explanation is that vertebral LVs strongly contribute to the reuptake of the CSF that is continuously produced by the ventricular choroid plexus and circulates in the ependymal canal of the spinal cord[8]. This model is supported by the presence of dura mater vLVs in the spine and their dense location around spinal nerve rami. A second and likely possibility is that the largest part of vLVs uptakes epidural fluids and cells. This assumption is supported by the presence of a large network of epidural vessels that extends from the peripheral lymphatic system (Fig. 5e, f) and drains the vertebral canal toward lymph nodes adjacent to the spinal cord (Fig. 6f, g).

Vertebral LVs localize mainly at the level of intervertebral ligaments or joints, much like cranial lymphatics that navigate in skull commissures alongside blood vasculature and spinal nerves[14]. We and others find that LVs avoid bone tissues[41,42]. Interestingly, the presence of LVs inside bone is observed in patients with vanishing bone disease (also called Gorham Stout disease GSD)[43]. GSD is a sporadic disease characterized by the presence of lymphatic vessels in bone and progressive bone loss. GSD can affect any bone in the body, but it most frequently affects the ribs and vertebrae, with poor prognosis[44,45]. Mechanisms preventing LV formation in bone are, however, currently unknown.

In contrast to skull sutures between skull cap bones, which are few, narrow and fixed, the vertebral disks, joints and ligaments between vertebral bones are numerous, large and mobile. They sustain the integrity and flexibility of the spine, which is predicted to require extensive interstitial fluid drainage. The large network of epidural vLV appears to be exquisitely adapted to this extensive drainage of non-neural peripheral tissues in the spine and to provide each vertebra with its proper clearance system toward collecting lymph node (Fig. 6g). It is predictable that defective

vLVs will alter vertebral and intervertebral tissue maintenance, leading to spine orthopedic pathologies.

The proximity of two distinct epidural and dura mater vLV circuits raises questions about the privileged immune status of the CNS. Like in the skull[13], a proximity between vLVs and CD45+ leukocytes is observed along the spine (Fig. 8). The spinal cord lymphatic vasculature thus appears as a potentially important immune surveillance interface between the CNS and peripheral tissues. The cervical, thoracic and lumbar regions directly drain into cervical, mediastinal and renal/lumbar lymph nodes, respectively, which suggests that the peri-lymphatic dendritic immune cells may transfer to those lymph nodes and initiate lymphocyte activation against specific pathogens or antigens. On the other hand, epidural and dura mater vLV may facilitate the propagation of peripheral infections through the vertebral canal toward neural tissues. For example, epidural vLVs may provide entry for meningitis infection into spinal meninges. The contact zone between DRGs and vLVs appears as another potential gate for entry into the CNS for pathogens. For example, bovine scrapie protein is first detected in the mesenteric lymph nodes and DRGs of lemurs or cattles infected orally with the agent of bovine spongiform encephalopathy (BSE)[46,47].

The spine is affected by variety of diseases including infections[47], acute spinal cord compressions[48], and degenerative spine disorders, a common condition in the ageing Western population[49]. Vertebral LVs are potential targets for these pathologies. The vertebral column is also a common site for skeletal metastatic tumors; as many as 70% of cancer patients have spinal metastases, and up to 10% of cancer patients develop metastatic cord compression[50]. Since lymphatics may serve as conduits for primary tumor cells in metastatic spreading[51], specific interference with the vertebral lymphatic vasculature could reduce or prevent spinal metastasis. Alternatively, lymphatic vessels are the first barrier for initiation of an adaptive immune response by antigen-presenting cells[52]. Facilitating the entry of immune cells into vLVs might thus also potentially improve the efficiency of immune checkpoint inhibitor treatments to destroy spinal tumor cells.

We found that adult vLVs rapidly expand in response to VEGF-C or tissue injury (Fig. 7c, f, i). In inflammatory conditions such as LPC-induced spinal cord demyelination, VEGF-C-pre-treatment resulted in a strong expansion of vLV circuits and the epidural immune-cell pool. These extra-parenchymal lympho-immune responses were associated with larger demyelinated lesions and increased number of peri-lesional neuronal cells in the parenchyme compared to LPC^control mice (Fig. 8g, h and Supplementary Fig. 7a). In this setting, the expansion of vLV coverage therefore exacerbated the cytotoxic inflammation and impaired the recovery of local tissue damage. In contrast to the beneficial effect of loss of skull meningeal lymphatics in EAE mice[17], LPC^Vegf-C trap and LPC-K14-VEGFR3-Ig mice failed to

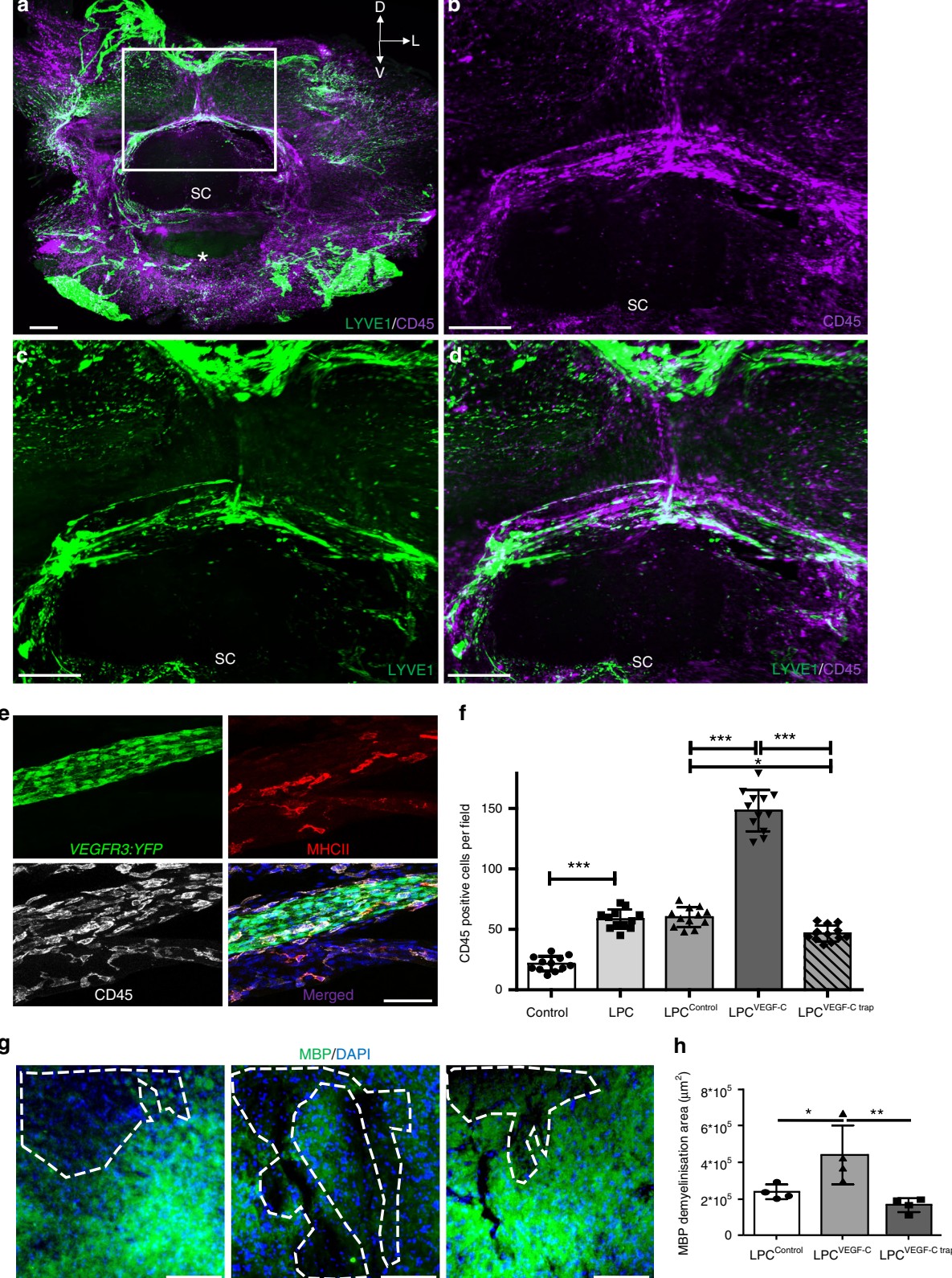

show significantly reduced immune-cell inflitration and demye-lination resulting from vLV blockade. This may be due to alter-native growth factors, incomplete vLV deletion, or simply harder to detect in a context of lower levels of inflammation in LPC-treated mice compared with EAE mice.

Vertebral LVs never contact the spinal cord tissue, even upon VEGF-C overexpression or acute spine lesion. In contrast, vLVs are closely apposed around the chains of sensory and sympathetic nervous ganglia. Although no lymphatic vascularization of sym-pathetic ganglia was observed, lymphatic vessels may provide

**Fig. 8** Interactions of spinal LVs with immune cells. **a–d** Double labeling of cleared cervical vertebral column segments with antibodies against LYVE1 (green) and CD45 (purple), spatial orientation (D: dorsal, L: lateral, V: ventral). **b–d** Magnified images of white box in (**a**). Merged images (**a**), (**d**) show CD45[+] leukocytes located along LYVE1[+] vLVs. White asterisk: vertebral ventral body; SC: Spinal cord. **e** Cryosection of a cervical vertebra from a *Vegfr3: YFP* mouse labeled with antibodies against MHCII (red) and CD45 (white). CD45[+] leukocytes including MHCII[+] antigen-presenting cells are located close to and inside a YFP[+] vLV (green) in the ligament flavum. **f** Quantification of CD45[+] cells in vertebral column whole-mount preparations (see stippled area in Fig. 7i). **g** Cryosections of the lumbar spinal cord from LPC-injured mice previously injected with AAV-VEGFR3$_{4-7}$-Ig (LPC$^{control}$), AAV-mVEGF-C (LPC$^{VEGF-C}$) or AAV-mVEGFR-3$_{1-3}$-Ig (LPC$^{VEGF-C trap}$) in the lumbo-sacral region. Images representative of the ipsilateral side showing MBP[+] myelin (green) and demyelinated area (dashed lines) with Hoechst[+] nuclear staining (blue) in (**g**). **h** Histograms showing quantification of MBP-negative demyelinated area (dotted line in (**g**)) at the lesion site. Demyelinated area is increased in LPC$^{VEGF-C}$ mice compared to LPC$^{control}$ mice. $n = 4$ biologically independent mice/independent experiment, and data represent mean$+/-$SD (error bar); one-way ANOVA with Tukey's multiple-comparisons test; *$p < 0.05$, ***$p < 0.001$. Source data are provided as a Source Data file. Scale bars: 300 μm (**a–d**); 50 μm (**e**); 100 μm (**g**)

molecular signals to the sympathetic neurons that control vascular tone of lymphatic ducts and cerebral arteries and arterioles. Previous observations also showed that adrenergic fibers connect to the thoracic lymphatic duct and also innervate the wall of lymph node arterioles[53,54]. The crosstalk between spine LVs and the sympathetic system is thus likely relevant for the regulation of peripheral lymph and glymphatic drainage and may coordinate them with the activity of brain and spine tissues. We speculate that a regulatory loop may link meningeal LV, sympathetic chain neurons and both CNS and peripheral fluid drainage.

To conclude, this study shows that the volume imaging technique allows the description of neurovascular systems by preserving the anatomy and the 3D-continuity of vascular and neural structures. In particular, our studies characterize the 3D-anatomical organization, the remodeling and function of the lymphatic vasculature along the spine. Our findings identify vertebral LVs as an important component for the maintenance and repair of vertebral tissues as well as a gatekeeper of CNS immunity.

## Methods

**Study approval and mice**. All in vivo procedures used in this study complied with all relevant ethical regulations for animal testing and research, in accordance to the European Community for experimental animal use guidelines (L358-86/609EEC). The study received ethical approval by the Ethical Committee of INSERM (n° 2016110111126651) and the Institutional Animal Care and Use Committee of ICM (Institut du Cerveau et de la Moelle épinière). Male C57BL/6J mice, *Vegfr3:YFP* lymphatic reporter mice[55], K14-VEGFR3-Ig mice[31], or *Prox1-eGFP* mice[50] between 2 and 3 months of age were used for all experiments.

**Tissue preparation**. Mice were given a lethal dose of Sodium Pentobarbital (Euthasol Vet) and perfusion-fixed through the left ventricle with 10 ml ice-cold PBS then 20 ml 4% paraformaldehyde (PFA) in PBS. To dissect the spine, the skin was completely removed, all the organs were eliminated and the ribs were removed to keep only the vertebral column from the cervical part until the lumbar part with the spinal cord inside. All the surrounding tissues including muscles, aorta and ligaments were maintained around the vertebral column. The spine was cut into pieces of about 0.5 cm (1–3 vertebrae) corresponding to the cervical, thoracic and lumbar regions. The different spinal segments were immediately immersed in ice-cold 4% PFA, fixed overnight at +4 °C, washed in PBS, and processed for staining.

**Sample pre-treatment in methanol for iDISCO[+] protocol**. We used a clearing protocol developed by Renier and colleagues, which is based on methanol dehydration and called the immunolabeling-enabled three-dimensional imaging of solvent-cleared organs (iDISCO[+], http://www.idisco.info)[21]. The steadily increasing methanol concentrations result in modest tissue-shrinkage (about 10%), while the "transparency" of tissues, such as the adult mouse brain, is increased. In detail, fixed samples were dehydrated progressively in methanol/PBS, 20, 40, 60, 80, and 100% for 1 h each (all steps were done with agitation). They were then incubated overnight in a solution of methanol 33%/dichloromethane 66% (DCM) (Sigma 270997-12X100 ML). After 2 × 1 h washes with methanol 100%, samples were bleached with 5% H$_2$O$_2$ in methanol (1 vol 30% H$_2$O$_2$/5 vol methanol) at 4 °C overnight. After bleaching, samples were rehydrated in methanol for 1 h each, 80%, 60%, 40%, 20%, and PBS. To clarify vertebral bone, we here added a decalcification step using Morse solution[23] during 30 min at RT. A weak acid treatment with Morse solution (1/1 tri-sodium citrate and 45% formic acid) decalcifies tissues efficiently while preserving their structure[56–58]. Samples were washed rapidly

with PBS then incubated 2 × 1 h in PTx2 (PBS/0.2% Triton X-100). At this step they were processed for immunostaining.

**Immunolabeling iDISCO[+] protocol**. Pretreated samples were incubated in PBS/ 0.2% Triton X-100/20% DMSO/0.3 M glycine at 37 °C for 24 h, then blocked in PBS/0.2% Triton X-100/10% DMSO/6% Donkey Serum at 37 °C for 24 h. Samples were incubated in primary antibody dilutions in PTwH (PBS/0.2% Tween-20 with 10 mg/ml heparin)/5% DMSO/3% Donkey Serum at 37 °C for 6 days. Samples were washed five times in PTwH until the next day, and then incubated in secondary antibody dilutions in PTwH/3% Donkey Serum at 37 °C for 4 days. Samples were finally washed in PTwH five times until the next day before clearing and imaging. We used the primary antibodies listed in the Supplementary Table 1. Primary antibodies were detected with the corresponding Alexa Fluor -555, -568, or -647 conjugated secondary antibodies from Jackson ImmunoResearch at 1/1000 dilution.

**iDISCO[+] tissue clearing**. After immunolabeling, samples were dehydrated progressively in methanol in PBS, 20, 40, 60, 80, and 100% each for 1 h. They were then incubated overnight in a solution of methanol 33%/DCM 66% followed by incubation in 100% DCM for 2 × 15 min to wash the methanol. Finally, samples were incubated in dibenzyl ether (DBE) (without shaking) until cleared (4 h) and then stored in DBE at room temperature before imaging.

**Paraffin section immunolabeling and imaging**. Vertebrae were decalcified for 3 weeks in 10% EDTA in 4% paraformaldehyde/PBS, dehydrated through ethanol, cleared in xylene and embedded in paraffin. Serial cross sections (5 μm thick) were immunostained with rabbit anti-mouse LYVE1 (1:100) polyclonal antibody (11-034, AngioBio Co). DAB (3,3′-Diaminobenzidine) staining was performed using the biotin avidin complex kit (PK-6100, Vectastain®Vector). Masson's trichrome staining was carried out using the Masson Trichrome Kit (BioGnost®- Ref. MST-100T). Hematoxylin (5 s) was used for counter staining. HRP-labeled paraffin sections were analyzed with a Zeiss Axio Scope.A1.

**Cryostat section immunolabeling**. For cryosections of the vertebral canal, fixed tissues underwent decalcification with 0.5 M EDTA, pH 7.4, at 4 °C. When the bone was becoming soft, samples were washed thoroughly with PBS and immersed in PBS containing 20% sucrose and 2% polyvinylpyrrolidone for 24 h at 4 °C, embedded in OCT compound (Tissue-Tek), and frozen at −80 °C. In total, 50–100-μm-thick sections were cut using a cryostat (Microm 550/CryoStar NX70; Thermo Fisher Scientific), air-dried, encircled with a pap pen, permeabilized with 0.3% PBS-TX, washed with PBS, and blocked in 5% donkey serum in PBS-TX at RT. After overnight primary antibody incubation at 4 °C in the same solution, the sections were washed with PBS and incubated with the appropriate fluorophore-conjugated secondary antibodies diluted in 0.3% PBS-TX for 1–2 h at RT. After washing with PBS, the sections were mounted with Vectashield mounting medium (Vector Laboratories), sealed with Cytoseal 60, and vertebra canal cryosections were imaged with a fluorescent macroscope.

For cryosections of the spinal cord, after fixation, the vertebral canal was opened and spinal cord dissected and dehydrated in a gradient of sucrose (10, 20, and 30% sucrose in PBS overnight for each solution at 4 °C). Then samples were embedded in OCT compound (Tissue-Tek), and frozen for storage at −80 °C. In total, 50–100-μm-thick sections were cut using a cryostat (Microm HM 550/ CryoStar NX70; Thermo Fisher Scientific), then blocked and permeabilized as free floating sections in TNBT (0.1 M Tris pH 7.4; NaCl 150 mM; 0.5% blocking reagent from Perkin Elmer; 0.5% Triton X-100) for 2 h at room temperature. Samples were incubated in primary antibodies diluted in TNBT overnight at 4 °C, washed five times in TNT (0.1 M Tris pH 7.4; NaCl 150 mM; 0.5% Triton X-100) and incubated with Alexa Fluor-conjugated secondary antibodies diluted in TNBT overnight at 4 °C. Finally, tissues were washed five times in TNT mounted in DAKO Fluorescent Mounting Media and spinal cord cryosections were imaged with a laser confocal microscope.

We used the primary antibodies listed in the Supplementary Table 1. Primary antibodies were detected with the corresponding Alexa Fluor -555, -568, or -647 conjugated secondary antibodies from Jackson ImmunoResearch at 1/1000 dilution.

**Whole-mount vertebral column immunolabeling.** After perfusion, lumbar segments corresponding to two adjacent vertebrae cranial and caudal to the site of LPC injection were harvested and fixed in 4% PFA overnight at 4 °C. The vertebral canal was opened laterally to expose dorsal and ventral sides and tissues were fixed in TNBT (0.1 M Tris pH 7.4; NaCl 150 mM; 0.5% blocking reagent from Perkin Elmer; 0.5% Triton X-100) for 2 h at room temperature before incubation with primary antibodies diluted in TNBT overnight at 4 °C. Whole mounts were washed five times with TNT (0.1 M Tris pH 7.4; NaCl 150 mM; 0.5% Triton X-100) and incubated with Alexa Fluor-conjugated secondary antibodies diluted in TNBT overnight at 4 °C. Finally, tissues were washed five times in TNT and imaged using a Leica DMIRB epifluorescence microscope using a ×4 objective.

For whole-mount staining of the larger vLV segments, fixed tissues were permeabilized with 0.3% Triton X-100 in PBS (PBS-TX) at room temperature (RT), then blocked with 5% donkey serum/2% bovine serum albumin/0.3% PBS-TX (DIM). Primary antibodies were diluted in DIM and samples were incubated in the primary antibody mix at least overnight at +4 °C. After washes with PBS-TX in RT, tissues were incubated with fluorophore-conjugated secondary antibodies in PBS-TX overnight at +4 °C, followed by washing in PBS-TX at RT. After post fixation in 1% PFA for 5 min, following washing with PBS, the stained samples were transferred to PBS containing 0.05% NaN₃ at +4 °C and imaged with a epifluorescence microscope and a ×4 objective.

We used the primary antibodies listed in the Supplementary Table 1. Primary antibodies were detected with the corresponding Alexa Fluor -555, -568, or -647 conjugated secondary antibodies from Jackson ImmunoResearch at 1/1000 dilution.

**LSFM, confocal and stereomicroscope imaging.** Cleared samples were imaged in transverse orientation with a LSFM (Ultramicroscope II, LaVision Biotec) equipped with a sCMOS camera (Andor Neo) and a 4 × /0.3 objective lens (LVMI-Fluor 4 × / 0.3 WD6, LaVision Biotec). Version v144 of the Imspector Microscope controller software was used. The microscope chamber was filled with DBE. We used single sided 3-sheet illumination configuration, with fixed x position (no dynamic focusing). The light sheet was generated by LED lasers (OBIS) tuned to 561 nm 100 mW and 639 nm 70 mW (LVBT Laser module 2nd generation). The light-sheet numerical aperture was set to 0.03. We used the following emission filters: 595/40 for Alexa Fluor-568 or -555, and -680/30 for Alexa Fluor-647. Stacks were acquired using 2 µm z steps and a 30 ms exposure time per step, with a Andor CMOS sNEO camera. The ×2 optical zoom was used for an effective magnification of (×8), 0.8 µm/pixel. Mosaic acquisitions were done with a 10% overlap on the full frame.

Laser scanning confocal micrographs of the fluorescently labeled cryosections were acquired using either a Zeiss Axioimager Z1 Apotome or a Leica TCS SP8 confocal microscope (air objectives ×10 Plan-Apochromat with NA 0.45 and ×25 Plan-Apochromat with NA 1.1) with multichannel scanning in-frame.

Fluorescent stereo micrographs were obtained with AxioZoom.V16 fluorescence stereo zoom microscope (Carl Zeiss) equipped with an ORCA-Flash 4.0 digital sCMOS camera (Hamamatsu Photonics) or an OptiMOS sCMOS camera (QImaging).

**Image processing and analysis.** For display purposes, a gamma correction of 1.47 was applied on the raw data obtained from the light-sheet fluorescent microscope.

Images acquired with Imspector acquisition software in tif fomat was converted with Imaris File Converter to IMS files. Mosaics were reconstructed with Imaris stitcher; then Imaris software (Bitplane, http://www.bitplane.com/imaris/imaris) was used to generate the orthogonal projections of data shown in all figures, perform area segmentation on a stack of image slices, and apply a color code to selected lymphatic networks.

**Tracer injections.** Thoracic-lumbar and lumbar-sacral spinal cord injection was performed in adult male and female C57BL/6J mice of 8–10-week of age. Mice were injected IP with Buprecare® solution and anaesthetized by Isoflurane gas (2–3%). The skin was incised at Th12-L1 (thoracic-lumbar injection) or L6-S1 (lumbar-sacral injection) vertebrae levels and paraspinal muscles covering the column were moved to the side. Dura mater and arachnoid membranes were incised using 30-gauge needle. Injections were then performed with a stereotaxic apparatus (Stoelting). Two or eight microliters of ovalbumin (2 mg/ml) or eight microliters of ovalbumin Alexa Fluor™ 555 Conjugate (OVA-A⁵⁵⁵; O34782, Invitrogen) or LYVE1 antibody was injected through a microcapillary (Glass Capillaries; GC120-15, Harvard Apparatus) connected to a Hamilton syringe. The microcapillary was introduced into one side of the spinal cord parenchyme. To avoid the release of OVA-A⁵⁵⁵; or LYVE1 Ab during the injection, a surgical glue was added to close the incision around the glass capillary. Injections were performed slowly (1 µl/min). Once injection was finished, the capillary was maintained for 2 min before retraction and a surgical

glue was added to close the hole made by the capillary, however some tracer leak always occurred despite these precautions. Tissue incisions were closed with Michel Suture Clips (7.5 × 1.75 mm; 12040-01, Fine Science Tool). After 15 or 45 min, mice were euthanized and perfused as described above in the "Tissue preparation" section.

**AAV injection.** Adult male mice were anesthetized with isoflurane (induction 4%, maintenance 2%) and placed in the stereotactic apparatus. AAVs serotype 9 were administered by either intra-cisterna magna (i.c.m.) injection or intra-lumbo-sacral (l.s.) injection. A single dose of 2 µl (10¹¹ viral particles per µl) of AAV-mVEGF-C, AAV-mVEGFR3₄–₇-Ig or AAV-mVEGFR-3₁–₃-Ig was administered into either C57BL/6J or K14-VEGFR3-Ig[55] male adult mice. i.c.m. and l.s. injections were performed using a Hamilton syringe with a 34-G needle and a flow rate of 0.5 µl/ min. The needle tip was retracted 2 min after the injection. Hundred microliters of 0.05 mg/kg of Buprecar® (buprenorphine) solution (intraperitoneal injection-IP) was used to relieve pre- and post-operative pain. All AAVs were produced by the vectorology platform of ICM.

**Spinal cord focal demyelination.** Adult male C57BL/6J mice were used. Lesions were induced in the thoracic-lumbar spinal cord by a stereotaxic injection of 1% of L-α-lysophosphatidylcholine (LPC) in PBS. Prior to the surgery, mice were anesthetized by intraperitoneal injection of ketamine (90 mg/kg) and xylazine (20 mg/kg) cocktail. Two longitudinal incisions into longissimus dorsi at each side of the vertebral column were performed, and the muscle tissue covering the column was moved to the side. Animals were placed in a stereotaxic frame, the 13th thoracic vertebra was fixed in between restraint bars designed for manipulations of mouse spinal cord (Stoelting, Wood Dale, IL), and intravertebral space was exposed by removing the connective tissue. The dura mater was perfored using a 30-gauge needle, and 1 µl NaCl or LPC was injected using a glass microcapillary (Glass Capillaries; GC120-15, Harvard Apparatus) attached via a connector to a Hamilton's syringe and mounted on a stereotaxic micromanipulator. Following injection, muscle sheaths were sutured with 3/0 Monocryl, and the skin incision was closed with 4/0 silk. After 7 days post injection, mice were perfused with 4% PFA in PBS; tissues were harvested and processed for iDISCO⁺ protocol or whole-mount and spinal cord stainings, as described above.

**Quantification and statistical analysis.** No statistical methods were used to predetermine sample size. Three to four mice were analyzed by experimental group ($n = 3$–4 mice/group). In vivo imaging quantification was performed with the Fiji software (ImageJ). The investigators were blinded during experiments and outcome assessment.

For quantification of vLV-associated immune cells, images of coronal sections of the cervical vertebral column labeled with immune-cell specific antibodies were acquired with a spinning-disk confocal (Nikon Eclipse Ti) using a ×20 objective. Cells were counted in the dorsal intervertebral spaces, in the vicinity of epidural vLVs ($n = 3$–4 mice/group, 3–4 fields/mouse). The number of CD45⁺ leukocytes was counted by surface unit (mm²). Myeloid cells (CD11b⁺), T lymphocytes (CD3e⁺), B lymphocytes (CD19⁺), and antigen-presenting cells (MHCII⁺) were counted as a percentage of CD45⁺ leukocytes.

For quantification of the vertebral lymphatic vessel diameter and area, images of whole-mount thoraco-lumbar vertebral canals stained with anti-LYVE1 antibody were acquired with a Leica DMIRB inverted epifluorescence microscope using a ×4 objective. The entire intervertebral space was quantified for each mice, and between 10 and 30 vessels were counted by mouse, depending on experimental conditions ($n = 4$ mice/group).

For quantification of demyelinated area, pictures of MBP-labeled tissues were acquired in the white mater, at the level of the LPC injection ($n = 4$ mice/group, 2 fields/mouse). Demyelinated areas were measured as surfaces of pixel intensity under 50 (scale 0–255).

For quantification of vLV-associated immune cells in mice with a demyelinating lesion, coronal sections of the cervical vertebral column were labeled with immune-cell specific antibodies and images were acquired with a Leica SP8 confocal using a ×25 objective. Cells were counted in the center of the demyelinated lesion and in the contralateral side of the spinal cord ($n = 4$ mice/ group, 2 fields/mouse).

Statistical data analysis was performed with the Prism 6.0 software (GraphPad). For continuous variables (area, diameter), data are presented as mean ± standard deviation (SD). For discrete variables (immune-cell number), data are presented as mean standard error of the mean (SEM). A two-tailed, unpaired Student's $t$ test or Mann–Whitney U test was done to determine statistical significance between two groups. For comparison between more than two groups, the one-way ANOVA test was performed, followed by Turkey's multiple comparison test. Differences were considered statistically significant if the $p$ value was <0.05 (*$p < 0.05$, **$p < 0.01$, ***$p < 0.005$, and ****$p < 0.0001$).

**Reporting summary.** Further information on research design is available in the Nature Research Reporting Summary linked to this article.

## Data availability
The source data underlying Fig. 7j, k and 8f–h as well as Supplementary Figs. 6b, d–f and 7a–f are provided as a Source Data file. All data supporting the findings of this study are available from the corresponding authors upon reasonable request.

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

## Acknowledgements

This work was supported by Institut National de la Sante et de la Recherche Medicale (to J.-L.T.), Agence Nationale Recherche (ANR-17-CE14-0005-03 to J.-L.T. and A.E.), Federation pour la Recherche sur le Cerveau (FRC 2017 to J.-L.T. and A.E.), Carnot Maturation (to L.J.), Leducq Foundation Transatlantic Network of Excellence (ATTRACT to A.E.), NIH (R01EB016629-01 to J.-L.T., NHLBI 1R01HLI125811, NEI 1R01EY025979-01, P30 EY026878 to A.E.) and the Yale School of Medicine (J.-L.T.). K.A.'s funding was provided by the Jane and Aatos Erkko Foundation, the European Research Council (European Union Horizon 2020 research and innovation program, grant 743155), the Wihuri Foundation, the Academy of Finland (Centre of Excellence Program 2014–2019, grants 271845 and 307366), and the Finnish Brain Foundation. We acknowledge the ICM-QUANT cellular imaging, ICM-histomics and -vectorology platforms. We are deeply grateful to Paul Muller and Daniel Mucida from the Rockefeller University for insightful exchanges on the preparation of decalcified samples.

## Author contributions

L.J. and J.-L.T. planned the project and designed experiments. L.J., J.d.B.N., L.S.B.B., L.H. M.G., T.M., S.A., B.B., Y.X., J.P., and M.-S.A. performed experiments. L.J. and J.d.B.N. generated all gain- and loss-of-vLV mouse models, as well as mice with focal LPC-induced demyelination, with the assistance of M.-S.A. H.N., and S.A. bred and provided K14-VEGFR3-Ig mice. E.S. and S.L. provided materials and methodological expertise. L.J., J.d.B.N., T.M., S.A., K.A., N.R., A.E., and J.-L.T. analyzed and interpreted data. J.-M.T. created art work of Figs. 1c, 2h, 5f, 6g, and 7a. L.J. and J.d.B.N. generated all other figures, with the assistance of L.S.B.B (Fig. 8e, Supplementary Figs. 4a and 6), L.H.M.G. and T.M. (Fig. 7j, k and 8f–h, Supplementary Fig. 7). L.J. generated all movies. J.-L.T. and A.E. wrote the paper. All authors edited the paper.

## Competing interests

The authors declare no competing interests.
