## [Peer Review File · Nature Communications]

Reviewers' comments:

Reviewer #1 - expert in lymphatic development (Remarks to the Author):

The authors investigate the spinal cord lymphatic network by using iDISCO tissue clearing and light sheet microscopy, which allows an unprecedented analysis of the system in 3D. They provide a very careful description of the spinal cord lymphatics, of the discrete subsets of linked but largely intervertebral lymphatics that they suggest provide drainage routes via epidural and extravertebral networks. Interesting observations include the higher density of lymphatics than previously reported in recent studies in skull and early studies in spinal cord, association of myeloid cells closely with lymphatics (as in skull), different patterning of lymphatic networks in different regions (cervical, thoracic, lumbar) of the spine and contacts between DRG lymphatics and sympathetic ganglia. They also go on to show that spinal cord lymphatics are VEGF-C responsive and that they are increased in number in inflammation following spinal cord injury, both observations that may have been expected. Overall, the study is well written and provides useful new anatomical information on these understudied vessel networks.

While the study uses elegant new imaging approaches to shed light on the anatomy of vertebral lymphatic networks, it remains exclusively descriptive using only one major approach. It lacks any definitive functional analysis of the physiological relevance of the lymphatic vessels and connections that it describes. For example, in the context of increased lymphangiogenesis in spinal cord injury, does this influence local inflammation? Do more or fewer lymphatics impact recovery of local tissue damage in this injury setting? In the absence of these networks are there pathological outcomes such as local fluid accumulation? Changes in myeloid populations? The current study would be vastly improved if it extended into such functional and important physiological questions and in its current state seems better suited for a more specialist journal.

Major considerations below:

- What is the physiological relevance of this beautifully described network in homeostasis or in a disease setting? Can the authors show for example the consequence of decreased or increased lymphatic networks in the spinal cord injury setting? Or in tissue fluid homeostasis or disease settings such as the many speculated upon in the discussion? Further detailed functional information in at least one such setting is needed for the study to provide more than descriptive anatomical information.
- The authors appear at points to suggest that this network contacts and may drain CSF (eg. in discussion - "the present imaging of extended vertebral lymphatic circuits that contact both peripheral lymph and CSF). Can the contacts between lymphatic vessels and CSF be directly

demonstrated using light sheet or a higher power/resolution approach such as EM? Current images are not sufficiently resolved to see such a direct contact and may indicate vessels imbedded in or on top of the membrane rather than penetrating and directly contacting and draining CSF. In the absence of active analysis of fluid drainage further evidence would be needed to make such claims.

- How does the iDISCO clearing agent affect membranes and the specific structures of the spinal cord? The lymphatics associated with for example the Dura - would these normally be in contact with both membranes but the tissue is disrupted and space between membranes increased due to processing? If there is a fixation artefact, validation of the major observations would require an independent method. To demonstrate that the anatomy is completely intact, it would be important to at least provide direct comparisons of a couple of different approaches in the tissue context being studied.

- Statements such as “Each vertebra is drained by semicircular dorsal and ventral vessels...” are misleading as no functional drainage data is provided in this current study that uses only fixed samples.

- The authors have nicely referenced the very early anatomical literature. However, the statement concluding that discussion section that “These predictions find support in the present imaging of extended vertebral lymphatic circuits that contact both peripheral lymph and CSF.” Doesn’t do some of the earlier studies justice. For example, in 1948 using rabbits and careful dye injections (lymphangiograms), Field and Brierly (J.Anat) traced the basic anatomy of spinal cord lymphatics including location along dorsal root ganglia and contact with the spinal cord membranes. They described functional, valveless lymphatics made up of broad endothelial cells that collected in sub-arachnoid space and carried fluid along major lymphatic channels as is suggested in this current paper. While the approaches and images used in such early anatomical studies were very simple, these anatomists were often impeccable and such studies go beyond “predictions” to careful anatomical description and correlation (similar to provided here but without modern technology).

- The authors refer to “plasticity of the lymphatic vasculature along the spine” the term plasticity is most often used to refer to plasticity between cell states (dynamic changes in cell identity) in lineages and so would be confusing in this context for many readers.

Reviewer #2 - expert in lymphatic development (Remarks to the Author):

The description of lymphatic vessels (LVs) in the meninges draining cerebrospinal fluids from head to neck areas in mice and humans (Louveau et al., 2015; Aspelund et al., 2015; Antila et al., 2017) is one of the prominent recent events in the field of lymphatic vascular biology. In this paper, Jacob et al. describe the LV network in the mouse vertebral column (vLVs). They confirm the finding by Antial and co-workers that the dense lymphatic vasculature is mostly located in the intervertebral spaces and further identify in thoracic vertebra a lymphatic network located dorsally and ventrally near the spinal cord (SC). They also describe the connectivity of LVs from different regions of vertebral column with lymph nodes and the thoracic duct. Intriguingly, LVs in the thoracic vertebra were found to be in close contact with sympathetic ganglia and located in dense CD45+ immune cell areas, suggesting an interaction between the lymphatic vascular system of the vertebra with the immune system as well as autonomous nervous system. The authors also show that vertebral lymphatics respond to VEGF-C/VEGFR-3 signaling.

The main positive point of the paper is that it describes the complete anatomical organisation of lymphatic vasculature around the spinal cord. The association of lymphatic vessels with sympathetic ganglia is very intriguing as well as perilymphatic accumulation of immune cells. Unfortunately, the study remains very descriptive and provides no insights into the potentially very important organ-specific function of this lymphatic vascular bed. The following additional experiments might be useful to clarify this question:

1. What changes are observed in vLVs during spinal injury and regeneration or in other disease models, such as EAE? Does lack of vertebral lymphatic vessels e.g. in *Krt14-Vegfr3-IgG* mice or vice versa enhancement of lymphangiogenesis with AAV VEGF-C, affect responses to spinal cord injury?
2. Does the molecular composition of vertebral LECs differ in any way from peripheral LECs?
3. What is the composition and origin of CD45+ immune infiltrates around vertebral LVs and what immune cells are trafficking via vLVs at steady state and inflammation? This can be investigated by staining for DCs, naïve and activated lymphocytes and macrophages.

Remark

On page 7, line 173-178. The authors describe the LVs in cervical and lumbar area. However, the figure referred to (S2, A-D) describe the LVs in cisterna magna and cervical areas. Results on lumbar LVs must be added to the Figure or the text must be corrected.

Minor remarks text:

1. page 2, line 21: central nervous system (CNS)
2. page 5, line 123: "Vertebral unit lymphatic architecture". The title needs to be rephrased
3. page 6, line 163: "red arrows" needs to be corrected by "salmon arrows" used in the figure legend

4. page 8, line 226: where did Figures 7K to 7N go?

Minor remarks figure legends:

1. page 14, line 399: no double arrow visible on the figure
2. page 15, line 414: dura mater: change D into DM, like in the Figure 3A.
3. page 15, line 427-428: LYVE1 is in green and CD45 in purple. This needs to be corrected.

Reviewer #3 - expert in tissue clearing and imaging methods (Remarks to the Author):

In this study, Jacob and colleagues investigate the anatomical 3-dimensional structure of the lymphatic vessels surrounding the spinal cord in particular in connection with inflammatory responses. A spinal cord injury is induced by injection of a physiological NaCl solution and a virus infection encoding VEGF-C (vascular endothelial growth factor C) was induced by injection of AAV into cisterna magnum, and the effects on the lymphatic vasculature was then evaluated by using the iDISCO clearing technique combined with Light sheet microscopy.

The images and movies are impressive. The work primarily characterizing the anatomy and therefore largely a descriptive study, with little or no quantification. This may be the first work of a kind, and therefore this is an important demonstration of the usefulness of 3D imaging, which has emerged over the last 5 years. I have only minor comments, except the issue that this study seem to be primarily an anatomical description, which is also reflected in the passiveness of the title "Anatomy of the vertebral column lymphatic network in mice", without a specific conclusion or hypothesis to test.

Specific comments

The authors use the original iDISCO protocol from Renier et al 2014, with some modifications, e.g. the decalcification step, since the specimen also has vertebrae bone structures. The authors mention the Morse solution, since it is not a part of the standard iDISCO protocol, the authors should provide a reference at this point.

Line 111:

“The overall vertebral lymphatic circuitry was surprisingly dense compared to the previously described lymphatic vasculature of skull meninges”

Q/C: There is no quantification of the lymphatic density. Is there a way to quantify the lymphatic density? If not, how does it make sense to compare with previous investigations? Perhaps the density could be quantified, which would be helpful for comparison in future studies.

Lines 172->

“LV patterning varies between cervical, thoracic and lumbar spine “

“More specifically, extra-vertebral LVs on the dorsal aspect of the spine were more abundant in lumbar compared to cervical vertebrae “

These are qualitative descriptions, does it make sense to write “more abundant” etc? Is it not possible to be more specific?

We thank the reviewers for the positive and constructive criticisms that we have addressed as detailed below in a point-by-point response. Reviewers' comments are copied verbatim in bold, our responses in normal font below.

Response to Reviewer #1 - expert in lymphatic development (Remarks to the Author):

The authors investigate the spinal cord lymphatic network by using iDISCO tissue clearing and light sheet microscopy, which allows an unprecedented analysis of the system in 3D. They provide a very careful description of the spinal cord lymphatics, of the discrete subsets of linked but largely intervertebral lymphatics that they suggest provide drainage routes via epidural and extravertebral networks. Interesting observations include the higher density of lymphatics than previously reported in recent studies in skull and early studies in spinal cord, association of myeloid cells closely with lymphatics (as in skull), different patterning of lymphatic networks in different regions (cervical, thoracic, lumbar) of the spine and contacts between DRG lymphatics and sympathetic ganglia. They also go on to show that spinal cord lymphatics are VEGF-C responsive and that they are increased in number in inflammation following spinal cord injury, both observations that may have been expected. Overall, the study is well written and provides useful new anatomical information on these understudied vessel networks.

We thank the reviewer for the positive comments on our work.

1. While the study uses elegant new imaging approaches to shed light on the anatomy of vertebral lymphatic networks, it remains exclusively descriptive using only one major approach. It lacks any definitive functional analysis of the physiological relevance of the lymphatic vessels and connections that it describes.

The revised paper now includes new data on vertebral lymphatic drainage and on spinal cord inflammation in gain-and loss-of VEGF-C signaling mice. The findings are illustrated in the revised manuscript by a new Figure 6, two modified Figures 7 and 8 and five new Supplementary Figures 2, 4, 5, 6 and 7. They are reported in three new paragraphs entitled 'vLV-mediated drainage of the vertebral column' (p. 8), 'Vertebral LVs respond to VEGF-C and spinal cord injury' (p. 8), and 'Effects of vLVs on myeloid and lymphoid cells' (p. 9). The title of the manuscript has also been changed to 'Anatomy and function of the vertebral column lymphatic network in mice'.

2. For example, in the context of increased lymphangiogenesis in spinal cord injury, does this influence local inflammation?

We addressed this point using LPC (lysophosphatidylcholine)-induced demyelinating lesions (Arnett et al. Science. 2004 Dec 17;306(5704):2111-5). Our findings are reported p. 9-10 in the paragraph 'Effects of vLVs on myeloid and lymphoid cells'. The new text is included below for the reviewer and is also highlighted in blue in the revised manuscript.

'On cryosections we observed leukocytes concentrated close to, or within, *Vegfr3-YFP*⁺ lymphatic vessels in inter-vertebral ligaments (**Fig. 8e**). CD45⁺ leukocytes included around 40% of C11b⁺ macrophages, 40% of CD3⁺ T cells, and 20% of CD19⁺ B cells. Furthermore, around 40% of CD45⁺ leukocytes expressed MHCII (**Supplementary Fig. 6a-d**).

Leukocyte numbers and ratios were similar between control, AAV-mVEGF-C and K14-VEGFR3-Ig homozygous and heterozygous mice (**Supplementary Fig. 6a-d**). In contrast, LPC-injury induced a strong increase in leukocyte numbers around vLVs that was further enhanced by AAV-mVEGF-C and reduced by mVEGF-C trap (**Fig. 8f**).

The size of demyelinated lesions, identified as spinal white matter areas devoid of MBP (Myelin Basic Protein) expression, was significantly increased in LPC^{VEGF-C} mice compared to control AAV treated mice (**Fig. 8g, h**). LPC^{VEGF-C trap} (AAV-mVEGF-C trap-treated) mice showed a significantly reduced lesion size when compared to LPC^{VEGF-C} (AAV-mVEGF-C-treated) mice, and a slight but not significant reduction in lesion area when compared to LPC^{control} mice (**Fig. 8g, h**).

Further quantifications were done on spinal cord sections in the lesioned area versus the contralateral uninjured side. As expected, LPC injection reduced the number of NeuN-

positive neurons in the peri-lesional area compared to the uninjured side (**Supplementary Fig. 7a**). Pre-treatment with control AAV-control or with AAV-VEGF-C trap had no effect compared to LPC injury alone, while AAV-mVEGF-C further reduced the number of NeuN positive neurons on the injured side, indicating deleterious effects of expanded vLVs on spinal cord demyelinating lesions (**Supplementary Fig. 7a**). LPC injection induced an increase in the number of Iba1⁺ microglia, F4/80⁺ macrophages and CD3⁺ T cells in the injected side compared to the contralateral side (**Supplementary Fig. 7b-d**). AAV-control had no effect on immune cell numbers, while the numbers of Iba1⁺, F4/80⁺ and CD3⁺ cells were further amplified in LPC^{VEGF-C} mice (**Supplementary Fig. 7b-d**). LPC^{mVEGF-C trap} mice showed significantly reduced leukocyte numbers when compared to LPC^{VEGF-C} mice, however leukocyte infiltration was not reduced when compared to untreated LPC^{control} mice (**Supplementary Fig. 7b-d**). Infiltration of CD3⁺ T cells and F4/80⁺ macrophages/microglia was also not significantly different between K14 homozygous and heterozygous LP- injured mice (**Supplementary Fig. 7e-f**). Altogether, these results demonstrate that a gain of vLVs amplified the cytotoxic effect resulting from LPC-induced injury.'

3. Do more or fewer lymphatics impact recovery of local tissue damage in this injury setting?

Revised Figure 7 now includes new data to address this important point. The revised text of this chapter (p.8-9 of the manuscript) now reads as follows:

Vertebral LVs respond to VEGF-C and spinal cord injury

To assess the dependence of vLVs on VEGF-C, we generated gain-of-VEGF-C signaling mice induced by either intra-cisterna magna (i.c.m.) or lumbo-sacral (l.s) injection of adeno-associated viral vectors (AAVs) encoding VEGF-C (AAV-mVEGF-C)¹⁵ (**Fig. 7a, d**). Control mice were injected with AAVs encoding soluble mVEGFR3₄₋₇-Ig (VEGFR3 ectodomains that do not bind VEGF-C) (AAV-control)³¹. One month later, mice were analyzed by Prox-1 immunostaining. Compared to controls, VEGF-C injected mice showed a strongly expanded vLV network, in particular of dorsolateral lymphatic rings in the intervertebral disk of cervical vertebrae after i.c.m. injection, (**Fig. 7b, c and movie 7**) and lumbar vertebrae after l.s. injection (**Fig. 7e, f**).

To determine if adult vLVs might respond to spinal cord injury, we injected LPC (1µl) into one side of the spinal cord at the thoraco-lumbar level (**Fig. 7g**). LPC is toxic to oligodendrocytes and rapidly induces demyelinating spinal cord lesions (**Fig. 7i** inset)^{22,32}. Within a week after the surgery, a robust extravertebral and intravertebral lymphangiogenesis was induced in LPC-injured mice (**Fig. 7h-i** and **movie 8**). To quantify the response, we opened the vertebral column to expose intravertebral vLVs that were stained with LYVE-1 on whole-mount preparations, followed by surface area measurements (stippled area in **Fig. 7i**). Pairwise Mann-Whitney U test comparison to control-AAV-injected LPC-injured mice (LPC^{control} mice) revealed a significant increase in vLV area after LPC injury (not shown), however this did not reach statistical significance in a multiple group comparison (**Fig. 7j**). LPC-induced lymphatic area was not affected by control AAV-mVEGFR3₄₋₇-Ig but significantly enhanced in mice pre-treated with AAV-mVEGF-C (LPC^{VEGF-C} mice) for one month (**Fig. 7j**). LPC injury in mice pre-treated by i.s. injection of AAVs encoding soluble mVEGFR3₁₋₃-Ig (mVEGF-C trap), (LPC^{VEGF-C trap} mice) for one month resulted in reduction of vLV area that was significantly different from LPC^{control} mice in a pairwise Mann-Whitney U-test comparison (not shown), but failed to reach statistical significance in a multiple group comparison (**Fig. 7j**). However, LPC injury in K14-VEGFR3-Ig homozygous mice, in which endogenous VEGF-C/VEGF-D ligands are constitutively trapped to prevent VEGFR3 signaling³¹, resulted in a significant reduction of vLV area compared to heterozygous littermates (**Fig. 7k**). Taken together, these data show that vLVs respond to VEGF-C and spinal cord injury. Furthermore, in mice with more vLVs, such as LPC^{VEGF-C} mice, we found larger demyelinated lesions and reduced number of peri-lesional cells after LPC injury, compared to LPC^{control} mice (**Fig. 8g, h**). In the setting of a spinal cord injury, the expansion of vLV coverage therefore exacerbates the cytotoxic inflammation and impairs the recovery of tissue damage.'

4. In the absence of these networks are there pathological outcomes such as local fluid accumulation? Changes in myeloid populations?

K14-VEGFR3-Ig homozygous mice were previously reported to have normal volume of brain fluids. Their brain water content (wet weight - dry weight)/wet weight) was similar to control mice (Aspelund A et al. J Exp Med. 2015 Jun 29;212(7):991-9). We thus assume that there is no fluid accumulation in the CNS of K14-VEGFR3-Ig homozygous mice.

We have analyzed K14-VEGFR3-Ig mice for the myeloid and lymphoid cell populations associated with vLVs, which had never been done before. This study is reported p. 9 and 10 in the chapter 'Effects of vLVs on myeloid and lymphoid cells' (see above response to comment 2).

We have also shown the negative impact of vLV growth on the spinal tissue damage induced LPC-lesion. See p.10 of the revised manuscript :

'The size of demyelinated lesions, identified as spinal white mater areas devoid of MBP (Myelin Basic Protein) expression, was significantly increased in LPC^{VEGF-C} mice compared to control AAV treated mice (**Fig. 8g, h**). LPC^{VEGF-C trap} mice showed a significantly reduced lesion size when compared to LPC^{VEGF-C} mice, and a slight but not significant reduction in lesion area when compared to LPC^{control} mice (**Fig. 8g, h**).

5. The current study would be vastly improved if it extended into such functional and important physiological questions and in its current state seems better suited for a more specialist journal.

We hope the reviewer will appreciate the set of functional and immunological studies added to our revised manuscript and consider it of high interest for the wide readership of *Nature Communications*. We note that we have developed several novel techniques to isolate, stain and functionally manipulate vertebral lymphatic vessels (see new **Fig. 6 and 7**) that should be very useful for future investigations of this poorly studied tissue. We also note the intense interest in meningeal lymphatic vessels sparked by their recent description in 2015. We believe our data significantly extend these descriptions to include the first state of the art analysis of lymphatics in the vertebral column.

Major considerations below:

6. What is the physiological relevance of this beautifully described network? The study should investigate the consequence of decreased or increased lymphatic networks in the spinal cord injury setting?

See response to comments 2 and 4 above.

7. Or in tissue fluid homeostasis or disease settings such as the many speculated upon in the discussion? Further detailed functional information in at least one such setting is needed for the study to provide more than descriptive anatomical information.

In addition to gain- and loss-of-VEGF-C signaling, analysis of spinal cord injury and inflammation responses, we have tested tissue fluid drainage in the vertebral column. The corresponding results are described on page 8 and copied below.

‘vLV-mediated drainage of the vertebral column

The function of the vLV system was explored by testing the drainage potential of epidural and dura mater vLVs. Molecular tracers were injected into one side of the spinal cord parenchyme at the thoraco-lumbar level, and their distribution around the injection site was examined 15 and 45 min after injection (a.i.) (**Fig. 6a**). We used as molecular tracers either LYVE-1 antibodies that were detected with a secondary antibody, or fluorescent albumin (OVA-A⁵⁵⁵). LSMF imaging of iDISCO-treated vertebral samples revealed that the injected markers localized in and around Prox-1⁺ vLVs of the epidural space and dura mater (white arrows, **Fig. 6b-e** and **movie 6**). Confocal imaging of decalcified and frozen samples showed that OVA-A⁵⁵⁵ localized within the vLV lumen (**Supplementary Fig. 5**), thus demonstrating the uptake and drainage properties of vLVs.

15 min after OVA-A⁵⁵⁵ injections into the thoraco-lumbar spinal cord, tracer accumulated in the ipsilateral paravertebral lymphatic vessel and mediastinal lymph node, in 9 out of 12 cases (**Fig. 6f**). Therefore, vLVs provide a regional outflow for epidural and subarachnoid fluids towards lymph nodes. A schematic model of the thoraco-lumbar lymphatic drainage circuitry is shown in **Fig. 6g**. ‘

8. The authors appear at points to suggest that this network contacts and may drain CSF (eg. in discussion - “the present imaging of extended vertebral lymphatic circuits that contact both peripheral lymph and CSF). Can the contacts between lymphatic vessels and CSF be directly demonstrated using light sheet or a higher power/resolution approach such as EM? Current images are not sufficiently resolved to see such a direct contact and may indicate vessels imbedded in or on top of the membrane rather than penetrating and directly

contacting and draining CSF. In the absence of active analysis of fluid drainage further evidence would be needed to make such claims.

We now provide additional images of vertebral meninges to localize their position with regard to the dura mater and the CSF. Three different approaches have been used and are illustrated in **Supplementary Fig. 4**. The corresponding results are reported on p. 7, 8 in & 'The vertebral lymphatic system includes epidural and dural vessels'. The text now reads as:

'Complementary examination of whole-mounted spinal cord meninges (Supplementary Fig. 4a) and cryo/paraffin sections (**Supplementary Fig. 4b-g**) confirmed that the dura mater lymphatic vessels were restricted around the DRGs and spinal nerves and located on the dorsal surface of the dura mater (**Supplementary Fig. 4c, f**), which is not in direct contact with the CSF'.

These two approaches allowed us to establish that DRG vLVs are attached to the outer aspect of the dura mater, but do not cross the dura mater and do not contact the CSF directly. We provide the reviewer with the following additional information on the techniques used:

1) Cervical vertebral meninges were dissected from adult *Vegfr3:YFP* mice (a lymphatic endothelial cell reporter model, Calvo CF et al. *Genes Dev.* 2011 Apr 15;25(8):831-44), then whole-mount meningeal preparations were stained with anti-LYVE-1 Ab and imaged with a Leica TCS SP8 confocal microscope (10x, mosaic of 16 pictures). LYVE-1⁺/YFP⁺ meningeal lymphatic vessels were only observed in the region of DRGs and spinal nerve rami.

2) Vertebral samples were decalcified, cryoprotected and frozen, then tissue cryosections (12 µm thick) were processed by IHC with anti-LYVE-1 Ab and imaged using a confocal laser scanning fluorescence microscope Olympus FLUOVIEW FV1000 (10X, 20X and 63X).

3) Vertebral samples were decalcified and dehydrated before inclusion in paraffin, then tissue sections (5 µm thick) were processed by IHC with anti-LYVE-1 Ab and imaged using a ZEISS AXIO Scope.A1 (10X, 20X et 63X).

9. In the absence of active analysis of fluid drainage further evidence would be needed to make such claims.

To assess vLV drainage potential, we have injected tracers into the spinal cord, as illustrated in **Fig. 6** and **Supplementary Fig. 5**. The results are reported p. 9 in & vLV-mediated drainage of the vertebral column' (see response to point 7 above)

10. How does the iDISCO clearing agent affect membranes and the specific structures of the spinal cord? The lymphatics associated with for example the Dura - would these normally be in contact with both membranes but the tissue is disrupted and space between membranes increased due to processing? If there is a fixation artefact, validation of the major observations would require an independent method. To demonstrate that the anatomy is completely intact, it would be important to at least provide direct comparisons of a couple of different approaches in the tissue context being studied.

We chose the IDISCO⁺ protocol because in contrast with other clearing methods, it is known to preserve the integrity of tissue structure and anatomy (Vigouroux RJ et al., Mol Brain. 2017 Jul 20;10(1):33). However, as suggested by the reviewer, we tested whether this assumption was correct. Corresponding results are reported p 6 in the paragraph 'Modular architecture of vertebral lymphatic vasculature'. The text now reads as:

'To verify that the IDISCO⁺ protocol preserved the integrity of tissue structure and anatomy²⁸, we performed additional immunostainings on decalcified EDTA-treated vertebral segments that were either cryoprotected and frozen, or dehydrated and embedded in paraffin. Confocal and conventional microscopy imaging of sections perfectly reproduced the 3D-images collected with a LSFM on IDISCO⁺ treated samples (**Supplementary Fig. 2**) and thus substantiated the IDISCO⁺ based-model described above. '

Furthermore, we dissected meninges from the vertebral canal to image separately the dura mater and the epidural space. Illustrations are shown in Supplementary Fig 4a and confirm the results from 3D-LSFM imaging of IDISCO-treated samples (Fig 5d-f).

Altogether, the results with different approaches are congruent and we therefore believe that they reflect the anatomy of the vertebral lymphatic system.

11. Statements such as “Each vertebra is drained by semicircular dorsal and ventral vessels...” are misleading as no functional drainage data is provided in this current study that uses only fixed samples.

We have now shown the drainage potential of vLVs and thus kept, in the second paragraph of the discussion, the sentence: ‘Each vertebra is drained by semicircular dorsal and ventral vessels, which exit the vertebral column at intervertebral foramina’.

12. The authors have nicely referenced the very early anatomical literature. However, the statement concluding that discussion section that “These predictions find support in the present imaging of extended vertebral lymphatic circuits that contact both peripheral lymph and CSF.” Doesn’t do some of the earlier studies justice. For example, in 1948 using rabbits and careful dye injections (lymphangiograms), Field and Brierly (J.Anat) traced the basic anatomy of spinal cord lymphatics including location along dorsal root ganglia and contact with the spinal cord membranes. They described functional, valveless lymphatics made up of broad endothelial cells that collected in sub-arachnoid space and carried fluid along major lymphatic channels as is suggested in this current paper. While the approaches and images used in such early anatomical studies were very simple, these anatomists were often impeccable and such studies go beyond “predictions” to careful anatomical description and correlation (similar to provided here but without modern technology).

Thanks for pointing this out. The sentence now reads:

‘We here extend these seminal findings by 3D-views of lymphatic vasculature organization and function in vertebral canal drainage (**Fig. 6h**)’

13. The authors refer to “plasticity of the lymphatic vasculature along the spine” the term plasticity is most often used to refer to plasticity between cell states (dynamic changes in cell identity) in lineages and so would be confusing in this context for many readers.

We replaced 'plasticity' by 'remodeling' (p 13, last paragraph of the discussion).

Response to Reviewer #2 - expert in lymphatic development (Remarks to the Author):

The description of lymphatic vessels (LVs) in the meninges draining cerebrospinal fluids from head to neck areas in mice and humans (Louveau et al., 2015; Aspelund et al., 2015; Antila et al., 2017) is one of the prominent recent events in the field of lymphatic vascular biology. In this paper, Jacob et al. describe the LV network in the mouse vertebral column (vLVs). They confirm the finding by Antila and co-workers that the dense lymphatic vasculature is mostly located in the intervertebral spaces and further identify in thoracic vertebra a lymphatic network located dorsally and ventrally near the spinal cord (SC). They also describe the connectivity of LVs from different regions of vertebral column with lymph nodes and the thoracic duct. Intriguingly, LVs in the thoracic vertebra were found to be in close contact with sympathetic ganglia and located in dense CD45+ immune cell areas, suggesting an interaction between the lymphatic vascular system of the vertebra with the immune system as well as autonomous nervous system. The authors also show that vertebral lymphatics respond to VEGF-C/VEGFR-3 signaling.

The main positive point of the paper is that it describes the complete anatomical organization of lymphatic vasculature around the spinal cord. The association of lymphatic vessels with sympathetic ganglia is very intriguing as well as perilymphatic accumulation of immune cells.

We thank the reviewer for the positive comments on our work.

Unfortunately, the study remains very descriptive and provides no insights into the potentially very important organ-specific function of this lymphatic vascular bed.

The revised paper now includes new data on vertebral lymphatic drainage and on spinal cord inflammation in gain-and loss-of VEGF-C signaling mice. The findings are illustrated in the revised manuscript by a new Figure 6, two modified Figures 7 and 8 and five new Supplementary Figures 2, 4, 5, 6 and 7. They are reported in three new paragraphs entitled 'vLV-mediated drainage of the vertebral column' (p. 8), 'Vertebral LVs respond to VEGF-C and spinal cord injury' (p. 8), and 'Effects of vLVs on myeloid and lymphoid cells' (p. 9). The title of the manuscript has also been changed to 'Anatomy and function of the vertebral column lymphatic network in mice'.

The following additional experiments might be useful to clarify this question:

1. What changes are observed in vLVs during spinal injury and regeneration or in other disease models, such as EAE? Does lack of vertebral lymphatic vessels e.g. in Krt14-Vegfr3-IgG mice or vice versa enhancement of lymphangiogenesis with AAV VEGF-C, affect responses to spinal cord injury?

The same comments were also raised by reviewer 1 and are addressed above (see points 2, 3 and 4).

2. Does the molecular composition of vertebral LECs differ in any way from peripheral LECs?

We agree with the reviewer that insights into the molecular properties of vLVs would be great to have. However, the extensive work required to perform functional studies on spine injury and vertebral fluid drainage did not allow us to add molecular analyses in the time frame allowed for revisions. In addition, it is conceivable that the molecular composition might not differ all that much from peripheral lymphatics, in which case not much will be learned from such labor-intensive and costly studies. We hope that the reviewer will understand this limitation.

We have nevertheless started to assess the CCL21 chemokine expression in vertebral lymphatics (See Fig) in control and gain-of-lymphatic mice. CCL21 ligand is expressed by

cranial LVs while its specific receptor CCR7 is expressed by neighboring immune cells in meninges, and CCL21/CCR7 signaling has been reported to enable meningeal immune cell traffic towards draining lymph nodes, especially in response to EAE (Louveau A et al. Nat Neurosci. 2018 Oct;21(10):1380-1391). In our future studies, these preliminary observations will be extended to LPC-injured mice and with quantitative analyses.

3. What is the composition and origin of CD45⁺ immune infiltrates around vertebral LVs and what immune cells are trafficking via vLVs at steady state and inflammation? This can be investigated by staining for DCs, naïve and activated lymphocytes and macrophages.

The same comment was raised by reviewer 1 and is addressed above (see response to point 2). Our findings are reported p. 9-10 in the paragraph 'Effects of vLVs on myeloid and lymphoid cells'.

Remarks

On page 7, line 173-178. The authors describe the LVs in cervical and lumbar area. However, the figure referred to (S2, A-D) describe the LVs in cisterna magna and cervical areas. Results on lumbar LVs must be added to the Figure or the text must be corrected.

We have corrected the **Supplementary Fig. 3a** and indicated the position of thoracic vertebrae.

Minor remarks text:

1. page 2, line 21: central nervous system (CNS).

We have edited the text and central nervous system (CNS) is introduced in the abstract.

2. page 5, line 123: "Vertebral unit lymphatic architecture". The title needs to be rephrased.

We have rephrased the title as ‘Modular architecture of vertebral lymphatic vasculature’

3. page 6, line 163: “red arrows” needs to be corrected by “salmon arrows” used in the figure legend.

We have edited the text.

4. page 8, line 226: where did Figures 7K to 7N go?

We have completely remodelled **Fig 7**. It is worth noting that, in the initial version of the manuscript, the picture of **Fig. 7H** was incorrectly attributed to a spinal cord injury by NaCl injection, while it had been taken from a LPC-injected mice. We apologize for this confusion. In the present revised manuscript, the legend of **Fig. 7i** correctly refers to a LPC-injured spinal cord.

Minor remarks on figure legends:

- 1. page 14, line 399: no double arrow visible on the figure**
- 2. page 15, line 414: dura mater: change D into DM, like in the Figure 3A.**
- 3. page 15, line 427-428: LYVE1 is in green and CD45 in purple. This needs to be corrected.**

All these remarks have been addressed according to the reviewer’s indications.

Response to Reviewer # 3 – expert in tissue clearing and imaging methods (Remarks to the Author):

In this study, Jacob and colleagues investigate the anatomical 3-dimensional structure of the lymphatic vessels surrounding the spinal cord in particular in connection with inflammatory responses. A spinal cord injury is induced by injection of a physiological NaCl solution and a virus infection encoding VEGF-C (vascular endothelial growth factor C) was induced by injection of AAV into cisterna magnum, and the effects on the lymphatic vasculature was then evaluated by using the iDISCO clearing technique combined with Light sheet microscopy.

The images and movies are impressive. The work primarily characterizing the anatomy and therefore largely a descriptive study, with little or no quantification. This may be the first work of a kind, and therefore this is an important demonstration of the usefulness of 3D imaging, which has emerged over the last 5 years.

We thank the reviewer for his positive comments on our work and agree with him that the 3D imaging is especially interesting with respect to vessels, and also axons, as it allows to follow, and thereby describe, their circuitry and to identify their connections with other circuits or anatomical structures. For example, the revised study reports for the first time a specific circuit between thoracolumbar vertebral lymphatics and mediastinal lymph nodes. It also provides the first evidence that vertebral lymphatics contact sympathetic ganglia.

I have only minor comments, except the issue that this study seem to be primarily an anatomical description, which is also reflected in the passiveness of the title “Anatomy of the vertebral column lymphatic network in mice”, without a specific conclusion or hypothesis to test.

The same issue was raised by reviewers 1 and 2 and is addressed above (see response to reviewer 1, points 2, 4 and 7). We hope the reviewer will appreciate the set of functional and immunological studies added to our revised manuscript and consider it of high interest for the wide readership of *Nature Communications*. We note that we have developed several novel techniques to isolate, stain and functionally manipulate vertebral lymphatic vessels (see **Fig. 6** and **7**) that should be very useful for future investigations of this poorly studied tissue. We also note the intense interest in meningeal lymphatic vessels sparked by their recent description in 2015. We believe our data significantly extend these descriptions to include the first state of the art analysis of lymphatics in the vertebral column.

Specific

comments:

The authors use the original iDISCO protocol from Renier et al 2014, with some modifications, e.g. the decalcification step, since the specimen also has vertebrae

bone structures. The authors mention the Morse solution, since it is not a part of the standard iDISCO protocol, the authors should provide a reference at this point.

We thank the reviewer for this suggestion. The use of the Morse solution (Morse A, 1945) for vertebral tissue decalcification is now underlined in the Results' p. 4 & 'Lymphatic vasculature pattern in the thoracic spine'. The text now reads as :

'To label vascular, immune and neural cell compartments within the intact vertebral column, segments of 2-4 vertebrae were dissected together with the surrounding muscle tissue and decalcified in Morse's solution²³.'

In M&M section, p. 24 & 'Sample pre-treatment in methanol for iDISCO⁺ protocol', we also indicate: 'A weak acid treatment with Morse solution (1/1 tri-sodium citrate and 45% formic acid) decalcifies tissues efficiently while preserving their structure and acting within a short time⁵⁶⁻⁵⁸'.

Line 111:

"The overall vertebral lymphatic circuitry was surprisingly dense compared to the previously described lymphatic vasculature of skull meninges"

We have deleted this sentence

Q/C: There is no quantification of the lymphatic density. Is there a way to quantify the lymphatic density? If not, how does it make sense to compare with previous investigations? Perhaps the density could be quantified, which would be helpful for comparison in future studies.

Quantifications of LV area have been added to **Fig.7**. We could not do those on iDISCO⁺-treated samples but have devised additional methods to quantify at least the intravertebral lymphatics (stippled area in revised **Fig. 7i**). See response to reviewer 1 above (point 2).

***Lines 172-: "LV patterning varies between cervical, thoracic and lumbar spine "
"More specifically, extra-vertebral LVs on the dorsal aspect of the spine were more abundant in lumbar compared to cervical vertebrae "***

Q/C: These are qualitative descriptions, does it make sense to write “more abundant” etc? Is it not possible to be more specific?

We agree with the reviewer and have revised the wording in p. 7 & ‘vLV patterning differs in the cervical, thoracic and lumbar spine’. The text now reads as:

‘Thoracic vertebral LVs were defined by a large dorsal extravertebral plexus (blue arrow, **Fig. 4c**) and a direct connection from ventrolateral DRG LVs (red arrow) to the thoracic lymphatic duct (green in **Fig. 4d**). The thoracic and lumbar regions appear to display similar extensions and patterns of extravertebral and intravertebral LVs (**Fig. 4c, e**).’

Reviewers' comments:

Reviewer #1 (Remarks to the Author):

The authors have provided a strong revision that includes extensive new data. The study represents a useful modern anatomical description of spinal cord lymphatics that also suggests interesting functions in pathological settings.

They show using an LPC model, that increased vLVs exacerbate demyelination in this setting, demonstrating a function in a pathological process. They describe the response of these lymphatic networks to VEGFC in some detail. This work is very nicely done. It significantly improves the impact of the study.

In addition, they have added significant further functional data looking at drainage and looking more closely at the dural lymphatics of the spinal column with alternative and complementary methods. Likewise, this new data improves the study.

Overall, this reviewer appreciates the effort made in revision.

The only issue that the authors should further address relates to more careful interpretation of functional data in Figure 5, 6 and Supp Fig 4:

- They show clearly in Supp Figure 4 that there is no contact between the CSF and the dural LVs. In Figure 6 they show drainage of tracer that was introduced into the spinal cord parenchyme by a direct injection method. This is a setting where they have punctured the arachnoid and allowed access of CSF tracer into the dural and epidural spaces from the puncture site. This caveat should be mentioned to not mislead the reader that these vessels drain the parenchyme in normal functional scenarios.

- The label in Fig 6g "CSF-dural LVs" is suggestive of a functional relationship that the data has not demonstrated (due to above issue).

- The statement on line 223 "...vLVs provide a regional outflow for epidural and subarachnoid fluids towards lymph nodes". Is quite likely only true in the presence of a punctured arachnoid and this caveat should be mentioned or the term sub-arachnoid removed.

- The suggestion made at line 317 that "...spinal meningeal lymphatic hotspots may also contribute to CSF uptake" is highly speculative based on the data and perhaps a more guarded statement should be considered.

Reviewer #2 (Remarks to the Author):

My main concern was the absence of functional analyses for this newly described lymphatic vascular bed. The authors now show that lymphatic vessels and immune infiltrates are increased in LPC injury model, and that this phenotype is exacerbated when mice are treated with VEGF-C. Thus, lymphatic vessels appear to play a detrimental role in this type of injury, which is an important results. Unfortunately, the immune cell characterization is done formally (more CD45+ cells in LPC injury + basic immune cell types in control vs AAV-VEGFC treated mice). The resolution of such analysis is not sufficient to allow a meaningful interpretation, as key CCR7+ and CXCR4+ populations that are expected to traffic via lymphatic vessels were not analysed. Therefore, it remains unclear which immune cell types accumulate when lymphatic vessels are expanded in the LPC model and how increased lymphatic vasculature can be detrimental. The authors should at least discuss various possibilities and the underlying mechanisms.

I have concerns regarding AAV-mVegfr3-Fc that was employed for VEGF-C blockade because it does not show any anti-lymphangiogenic activity. This is somewhat surprising because the authors repeatedly state that vertebral lymphatics are VEGF-C dependent. How do the authors know that there was indeed production of functional Vegfr3-Fc protein? Were other VEGFR3-sensitvive lymphatic affected in the same mice? If not, it remains possible that the batch of virus used was defective and the results are not reliable.

Minor comment:

P 10 "in contrast, LPC injury induced a strong increase in leukocyte numbers around vLVs that was further enhanced by AAV-mVEGFC and reduced by mVEGF-trap". I agree with the enhancement part, but the trap, as discussed above, has no effect. Similarly, in other places the comparisons of VEGF-C trap data to VEGF-C overexpression data are meaningless, they should be compared to AAV-control.

Reviewer #3 (Remarks to the Author):

I think the authors have responded well to my concerns. I have no further comments or questions.

NCOMMS-18-24495A
Response to reviewers

We thank the reviewers for the positive and constructive criticisms that we have addressed, as detailed below in a point-by-point response. Reviewers' comments are copied verbatim in bold, our responses in normal font below.

Response to Reviewer #1

The authors have provided a strong revision that includes extensive new data. The study represents a useful modern anatomical description of spinal cord lymphatics that also suggests interesting functions in pathological settings.

They show using an LPC model, that increased vLVs exacerbate demyelination in this setting, demonstrating a function in a pathological process. They describe the response of these lymphatic networks to VEGFC in some detail. This work is very nicely done. It significantly improves the impact of the study. In addition, they have added significant further functional data looking at drainage and looking more closely at the dural lymphatics of the spinal column with alternative and complementary methods. Likewise, this new data improves the study.

Overall, this reviewer appreciates the effort made in revision.

The only issue that the authors should further address relates to more careful interpretation of functional data in Figure 5, 6 and Supp Fig 4:

- They show clearly in Supp Figure 4 that there is no contact between the CSF and the dural LVs. In Figure 6 they show drainage of tracer that was introduced into the spinal cord parenchyme by a direct injection method. This is a setting where they have punctured the arachnoid and allowed access of CSF tracer into the dural and epidural spaces from the puncture site. This caveat should be mentioned to not mislead the reader that these vessels drain the parenchyme in normal functional scenarios.

We thank the reviewer for this accurate comment and have add the following sentence to the & 'vLV-mediated drainage of the vertebral column':

'It is worth noting that this surgery procedure punctures the dura mater, which allows access of injected tracer into the epidural space located above spinal meninges, locally at the puncture site' (3rd sentence).

- The label in Fig 6g "CSF-dural LVs" is suggestive of a functional relationship that the data has not demonstrated (due to above issue).

We have replaced 'CSF-dural LVs' by 'CSF & dura mater' in Fig. 6g

- The statement on line 223 "...vLVs provide a regional outflow for epidural and subarachnoid fluids towards lymph nodes". Is quite likely only true in the presence of a

punctured arachnoid and this caveat should be mentioned or the term sub-arachnoid removed.

We have removed the term sub-arachnoid in the sentence on line 223.

- The suggestion made at line 317 that “....spinal meningeal lymphatic hotspots may also contribute to CSF uptake” is highly speculative based on the data and perhaps a more guarded statement should be considered.

We thank the reviewer for his careful reading and have modified the sentence at line 317 as follows:

‘Subsequent surgeries specifically delivering fluorescent tracers within the spinal cord subarachnoid space will be required to determine whether the vertebral dural vLVs may be hotspots contributing to CSF uptake’.

Response to Reviewer #2

My main concern was the absence of functional analyses for this newly described lymphatic vascular bed. The authors now show that lymphatic vessels and immune infiltrates are increased in LPC injury model, and that this phenotype is exacerbated when mice are treated with VEGF-C. Thus, lymphatic vessels appear to play a detrimental role in this type of injury, which is an important results.

Unfortunately, the immune cell characterization is done formally (more CD45+ cells in LPC injury + basic immune cell types in control vs AAV-VEGFC treated mice). The resolution of such analysis is not sufficient to allow a meaningful interpretation, as key CCR7+ and CXCR4+ populations that are expected to traffic via lymphatic vessels were not analysed. Therefore, it remains unclear which immune cell types accumulate when lymphatic vessels are expanded in the LPC model and how increased lymphatic vasculature can be detrimental. The authors should at least discuss various possibilities and the underlying mechanisms.

We agree with the reviewer that the activation and differentiation profiles of immune cells associated with vLVs are crucial to understanding the activation and trafficking of antigen-presenting cells towards draining lymph nodes. However, the low number of immune cells collected from the spinal cord, meninges and epidural spaces precludes flow cytometry analyses of immune cell phenotypes. This makes tedious IHC and expensive RNA-Seq approaches necessary to achieve the extent of phenotypic characterization requested by the reviewer. Such experiments are in progress in our lab but will require considerable technical work and financial efforts, which are beyond the scope of the present study and report.

I have concerns regarding AAV-mVegfr3-Fc that was employed for VEGF-C blockade because it does not show any anti-lymphangiogenic activity. This is somewhat surprising because the authors repeatedly state that vertebral lymphatics are VEGF-C

dependent. How do the authors know that there was indeed production of functional Vegfr3-Fc protein? Were other VEGFR3-sensitve lymphatic affected in the same mice? If not, it remains possible that the batch of virus used was defective and the results are not reliable.

We thank the reviewer for his critical comment and are happy to provide him with new data and explanations on loss-of-vLV models.

Fig. 1 Ventral view of vertebral lymphatic vessels in a whole-mount preparation of thoraco-lumbar vertebrae after removal of the spinal cord. a-c Labeling with anti-LYVE-1 antibody (white) shows the lymphatic vessel pattern in WT mice (a) and K14-VEGFR3-Ig hom. mice (b). Note that the vertebral lymphatic vasculature is reduced but not completely ablated in VEGF-C trap mice, where lymphatic vessels persist near dorsal root ganglia (DRG, red arrows). After focal LPC of the spinal cord, K14-VEGFR3-Ig hom. mice displayed growth of remaining vertebral lymphatic vessels (blue arrows) (c).

1. First, we maintain the statement that vertebral lymphatics are VEGF-C-dependent because AAV-VEGF-C strongly increases their growth (revised manuscript, **Fig. 7d-f**), and they are strongly reduced in K14-Vegfr3-Fc mice (Figures for reviewers, **Fig. 1a, b**). The presence of a few lymphatic vessels in the vertebral meninges of K14-Vegfr3-Fc mice nevertheless suggests that in addition to VEGF-C signaling, vertebral lymphatic growth might involve alternative regulation that remains to be characterized.

2. We know that:

- there is efficient production of VEGFR3-Ig, as previously reported by Antila S. et al. (Figures for reviewers, **Fig. 2**).

Fig. 2 Detection of VEGFR3-Ig after AAV-VEGFR3-Ig delivery. Adult WT mice (8 week-old) were injected intraperitoneally with AAV-VEGFR3-Ig. At 1, 2 and 5 weeks after injection, the proteins encoded by the viral vector were detected in serum by western blotting. mVEGFR3-Ig was detected already 24 h after AAV injection (**Supplemental Fig. 7F**, Antila S. et al. *J Exp Med.* 2017 Dec 4;214(12):3645-3667).

- the same batch of AAV-VEGFR3-Ig virus is highly efficient in other contexts, and we have just generated loss-of-meningeal lymphatic mice for further implantation with intracerebral glioblastoma cells (Figures for reviewers, **Fig. 3a**). This loss-of-cranial lymphatic model showed a significant reduction in survival following tumor implantation, in contrast with gain-of-cranial lymphatic models, which rejected the tumor graft. These data are confidential and included in a manuscript in revision at Nature (Song et al.).

- in the same animal, intra-cisternal administration of AAV-VEGFR3-Ig leads to a nearly complete loss of the lymphatic vasculature in the cranial meninges, a highly VEGF-C sensitive vascular structure, while cervical lymphatic vessels were only mildly and non-significantly reduced (Figures for reviewers, **Fig. 3b**). We assume that the predominant location of vertebral lymphatics, in the epidural space above the dura mater, limits their access to sufficient VEGFR3-Ig from the CSF, and prevents them from trapping endogenous VEGF-C. We have now clarified this better in the manuscript.

Fig. 3 Intra-cisterna magna injection of AAV-VEGFR3-Ig (sVEGFR3). **a** C57BL/6 mice were injected with AAV-CTRL or AAV-sVEGFR3 into the cisterna magna. Mice were euthanized and the dura was collected to image the lymphatic vasculature (LYVE-1) in the confluence of sinuses ($n = 5$). Mean \pm SD, Mann Whitney U test $*P < 0,05$, $***P < 0,001$. **b** A C57BL/6 mouse was injected with AAV-sVEGFR3 into the cisterna magna and examined after 4 weeks to assess alteration of the cranial and cervical lymphatic vasculature. Note that cranial lymphatic vessels were almost completely ablated in the confluence of sinuses (blue arrowheads; compare with AAV-CTRL in (a)) while some cervical lymphatic vessels persisted both in the epidural space and close to the DRG (red arrows).

We are currently testing alternative surgical approaches. to deliver AAV-VEGFR3-Ig into the epidural space, under the ligamentum flavum and above the dura mater, but the establishment of these techniques is technically complex and in active development.

3. We have shown that, although VEGF-C trap did not significantly decrease the vertebral lymphatic coverage (Figures for reviewers, **Fig. 4a**), it efficiently reduced the caliber of vertebral lymphatic vessels (Figures for reviewers, **Fig. 4b**). This information was not included in our revised manuscript due our decision to make the gain-of-function model in the context of spinal inflammation the primary focus of the study. In the present revised manuscript, it has been substituted to **Fig. 7j, k** (lymphatic coverage), which are now shown as **Supplementary Fig. 6e, f**.

Fig. 4 Quantitative analysis of vertebral lymphatic vasculature in loss-of-mVEGF-C signaling mice.

a Quantification of lymphatic vessel diameter area after LPC-spinal cord injury in gain- and loss-of-mVEGF-C signaling mice. Quantification of lymphatic vessel diameter area in LPC-injured K14-VEGFR3-Ig hom. mice and -K14-VEGFR3-Ig het. (control) mice. Data in (a) show mean + /- SD; Data in e show mean + /- SD; one-way ANOVA with Tukey's multiple-comparisons test), data in f show mean + /- SD, Mann Whitney U test *P<0,05, ***P<0,001. **b** Quantification of lymphatic vessel diameter after LPC-spinal cord injury in gain- and loss-of-mVEGF-C signaling mice. Quantification of lymphatic vessel diameter in LPC-injured K14-VEGFR3-Ig hom. mice and -K14-VEGFR3-Ig het. (control) mice. Data in (b) show mean + /- SD; one-way ANOVA with Tukey's multiple-comparisons test), data in k show mean + /- SD, Mann Whitney U test *P<0,05, ***P<0,001.

We hope that these new data will convince the reviewer that the AAV-VEGF-C trap approach allows the manipulation of vertebral lymphatic vessels and that it contributes novel and important information on vertebral lymphatic function. One reason why VEGF-C has a stronger effect on the lymphatic vessels may be its significantly larger size and the restricted access and distribution of the VEGFR3-Ig protein. In addition, there might be a difference in the growth versus survival response of vertebral lymphatic endothelial cells to VEGF-C. A small increase of VEGF-C above physiological level may stimulate lymphatic vessel growth, while a regression of lymphatic vessels may require a larger reduction of VEGF-C below the physiological level.

4. As a complete loss-of-vertebral lymphatics could not be achieved with the AAV-VEGFR3-Ig approach, we also used the K14-VEGFR3-Ig mouse model to explore the impact of a lack of vertebral lymphatics on spinal cord inflammation. Although the vertebral lymphatics were almost completely absent from K14-VEGFR3-Ig mice, the lymphatic vasculature was partially reconstituted within a few days following spinal cord lesion (Figure for reviewers, **Fig. 1c**), presumably resulting from the remaining lymphatic vessels/lymphatic endothelial cells upon stimulation by VEGF-C produced by macrophages that are recruited around the lesion site. This response partially restored the local vertebral lymphatic circuitry.

Minor comment:

P 10 “in contrast, LPC injury induced a strong increase in leukocyte numbers around vLVs that was further enhanced by AAV-mVEGFC and reduced by mVEGF-trap”. I agree with the enhancement part, but the trap, as discussed above, has no effect. Similarly, in other places the comparisons of VEGF-C trap data to VEGF-C overexpression data are meaningless, they should be compared to AAV-control.

The diameter of vLV is significantly reduced in AAV-VEGFR3-Ig treated mice (revised manuscript, **Fig 7j, k**). This significant alteration can be correlated with a reduction in the leukocyte population (CD45⁺ cells) associated with vLVs in the epidural space (revised manuscript, **Fig. 8f**). We have thus conserved, p.10, the sentence ‘in contrast, LPC injury induced a strong increase in leukocyte numbers around LVs that was further enhanced by AAV-mVEGFC and reduced by mVEGF-trap’.

Reviewer #3

I think the authors have responded well to my concerns. I have no further comments or questions.

REVIEWERS' COMMENTS:

Reviewer #2 (Remarks to the Author):

The authors have fully addressed my concerns about the activity of VEGF-C trap.